   

# Pro-inflammatory macrophage activation does not require inhibition of oxidative phosphorylation

Andréa B Ball [1], Anthony E Jones [1], Kaitlyn B Nguyễn[1], Amy Rios[1], Nico Marx [2], Wei Yuan Hsieh[3], Krista Yang[1], Brandon R Desousa [1], Kristen K O Kim [1], Michaela Veliova[1], Zena Marie del Mundo [3], Orian S Shirihai [4], Cristiane Benincá [4], Linsey Stiles[1,4], Steven J Bensinger [1,3] & Ajit S Divakaruni [1 ✉]

## Abstract

Pro-inflammatory macrophage activation is a hallmark example of how mitochondria serve as signaling organelles. Oxidative phosphorylation sharply decreases upon classical macrophage activation, as mitochondria are thought to shift from ATP production towards accumulating signals that amplify effector function. However, evidence is conflicting regarding whether this collapse in respiration is essential or dispensable. Here we systematically examine this question and show that reduced oxidative phosphorylation is not required for pro-inflammatory macrophage activation. Different pro-inflammatory stimuli elicit varying effects on bioenergetic parameters, and pharmacologic and genetic models of electron transport chain inhibition show no causative link between respiration and macrophage activation. Furthermore, the signaling metabolites succinate and itaconate can accumulate independently of characteristic breaks in the TCA cycle in mouse and human macrophages, and peritoneal macrophages can be activated in vivo without inhibition of oxidative phosphorylation. The results indicate there is plasticity in the metabolic phenotypes that can support pro-inflammatory macrophage activation.

**Keywords** Macrophage Polarization; Oxidative Phosphorylation; Mitochondrial Signaling; Immunometabolism; Itaconate
**Subject Categories** Immunology; Metabolism

## Introduction

Metabolic alterations are tightly linked to macrophage function and fate (O'Neill and Pearce, 2016; Van den Bossche et al, 2017; Wang et al, 2019; Garaude et al, 2016). Classical, pro-inflammatory activation triggered by exposure to lipopolysaccharide (LPS) causes a dramatic shift in macrophage energy metabolism: ATP production is shifted almost entirely to glycolysis, the TCA cycle is rewired, and oxidative phosphorylation is substantially inhibited (Jha et al, 2015; Tannahill et al, 2013; Van den Bossche et al, 2017).

It is generally accepted that this respiratory inhibition and a switch to "aerobic glycolysis"—largely due to excessive nitric oxide production (Bailey et al, 2019; Van den Bossche et al, 2016; Palmieri et al, 2020a)—is an essential feature of pro-inflammatory activation (Palmer, 2022; Fitzgerald and Kagan, 2020; Ryan and O'Neill, 2020). Conventionally, mitochondria are thought to be repurposed away from oxidative phosphorylation in order to generate metabolites and other mitochondrial signals that enhance macrophage function (Jones and Divakaruni, 2020; Mills et al, 2016; Lampropoulou et al, 2016; Ryan et al, 2019). Genetic loss-of-function studies may also imply a specific role for oxidative phosphorylation and respiratory chain function: myeloid-specific loss of a subunit of respiratory complex I enhances the pro-inflammatory phenotype whereas no phenotypic changes were observed upon myeloid-specific ablation of the mitochondrial pyruvate carrier (Cai et al, 2023; Ran et al, 2023).

Other reports, however, have demonstrated preserved or even enhanced pro-inflammatory macrophage function under conditions where oxidative phosphorylation remains functional (Cao et al, 2020; Bailey et al, 2019; Palmieri et al, 2020a; Vijayan et al, 2019). As has been previously suggested, physiologically relevant mitochondrial signals such as redox changes, metabolite accumulation, or superoxide production from reverse electron transport (RET) do not require mitochondrial damage or dysfunction (Robb et al, 2018; Buescher et al, 2015). It, therefore, remains unclear to what extent the LPS-induced collapse in mitochondrial ATP production is a requisite, causal driver of macrophage effector function or simply an associated epiphenomenon.

Here we use pharmacologic, genetic, human, and in vivo models to systematically detail that this well-established reduction in oxidative phosphorylation is unexpectedly dispensable for the induction of the macrophage pro-inflammatory response. We show that (i) not all pro-inflammatory stimuli decrease mitochondrial respiration, (ii) the signaling metabolites itaconate and succinate can accumulate without characteristic 'breaks' in the TCA cycle in mouse and human macrophages, (iii) pharmacologic and genetic inhibition of the respiratory chain does not amplify the pro-

[1]Department of Molecular and Medical Pharmacology, University of California, Los Angeles, Los Angeles, CA, USA. [2]Institute of Integrative Cell Biology and Physiology, Bioenergetics and Mitochondrial Dynamics Section, University of Münster, Schloßplatz 5, D-49078 Münster, Germany. [3]Department of Microbiology, Immunology, and Molecular Genetics, University of California, Los Angeles, Los Angeles, CA, USA. [4]Department of Medicine, University of California, Los Angeles, Los Angeles, CA, USA. ✉E-mail: adivakaruni@mednet.ucla.edu

inflammatory response, (iv) respiratory inhibition does not temporally align with the induction of the pro-inflammatory response, and (v) mouse peritoneal macrophages activated in vivo via sterile inflammation retain normal oxidative phosphorylation. The results suggest our conventional model of metabolic repurposing of macrophage activation is incomplete, and putative mitochondrial signals that enhance pro-inflammatory macrophage effector function are compatible with robust oxidative phosphorylation.

## Results

### Not all pro-inflammatory stimuli elicit uniform bioenergetic responses

LPS is an outer membrane component of Gram-negative bacteria and frequently used as a stimulus for classical, pro-inflammatory macrophage activation. LPS is a Toll-like receptor 4 (TLR4) agonist, and when offered at high concentrations, it can activate both the myeloid differentiation primary response 88 (MyD88) and the TIR-domain-containing adapter-inducing interferon-β (TRIF) adapter proteins (Fig. 1A; Appendix Fig. S1A) (Akira and Takeda, 2004). Treatment of mouse bone marrow-derived macrophages (BMDMs) with 50 ng/mL LPS for 24 h resulted in decreased mitochondrial respiration, as is well established (Fig. 1B,C) (Bailey et al, 2019; Mills et al, 2016; Van den Bossche et al, 2016; Huang et al, 2014). Since TLR4 is upstream of both MyD88 and TRIF, we sought to determine which signaling arm is responsible for the decrease in respiration. We exposed BMDMs to Pam3CSK4 (Pam3; a TLR2 agonist which specifically activates MyD88) as well as polyinosinic-polycytidylic acid (Poly I:C; a TLR3 agonist which specifically activates TRIF) (Fig. 1D) (Akira and Takeda, 2004; O'Neill et al, 2013). Treatment with either Pam3 or Poly I:C for 24 h was sufficient to elicit the expression of characteristic pro-inflammatory genes and the secretion of inflammatory cytokines (Appendix Fig. S1B–D). However, neither Pam3 nor Poly I:C caused the same profound respiratory inhibition observed with LPS (Fig. 1E,F).

We, therefore, hypothesized that engagement of both MyD88 and TRIF was required for substantial loss of mitochondrial respiration. Indeed, co-treatment with both ligands was sufficient to decrease mitochondrial respiration after 24 h (Fig. 1E,F). This relationship was also reproduced with Poly I:C and imiquimod (IMQ), a TLR7 agonist that engages MyD88 (Appendix Fig. S1E,F). The MyD88-linked stimuli Pam3 and IMQ slightly decreased maximal respiration, though not to the same extent as when used in combination with Poly I:C (Fig. 1F; Appendix Fig. S1E). Furthermore, treatment with heat-killed *Staphylococcus aureus* (HKSA)—a physiologically relevant TLR2 agonist—also did not decrease ATP-linked respiration and caused only a minimal defect in maximal respiration (Appendix Fig. S1G).

We then used other approaches, including measurements of mitochondrial morphology, mitochondrial membrane potential, and NADH fluorescence lifetime imaging (Jones et al, 2021) to determine whether this cooperativity between Pam3 and Poly I:C extended to other aspects of mitochondrial function. As with respirometry, neither ligand alone appreciably altered mitochondrial morphology after 24 h. However, co-treatment with both Pam3 and Poly I:C decreased the aspect ratio, suggesting increased fragmentation (Fig. 1G; Appendix Fig. S1H). Unexpectedly, we did not observe the same cooperativity

with measurements of either mitochondrial membrane potential or NADH fluorescence lifetime imaging microscopy (FLIM). Treatment with Pam3 alone for 24 h resulted in increased membrane potential relative to vehicle controls, while Poly I:C alone and the combination of both ligands did not cause a change relative to vehicle controls (Fig. 1H,I). To control for changes in mitochondrial content, mitochondrial membrane potential was measured as the average tetramethylrhodamine, ethyl ester (TMRE) intensity in areas positive for MitoTracker Green (MTG) staining. Additionally, FLIM analysis of the total cellular NADH pool revealed a heterogenous response in BMDMs treated with Pam3, Poly I:C, or both ligands for 24 h (Fig. 1J,K). The most pronounced difference observed was a shorter lifetime in Pam3-treated macrophages relative to vehicle controls, and in total, the results indicate the redox status and/or size of the macrophage pyridine nucleotide pool are subject to signal-specific remodeling during pro-inflammatory activation (Song et al, 2024; Schaefer et al, 2019). Collectively, these results show the bioenergetic response during classical macrophage activation is not uniform, and engagement of both the MyD88 and TRIF adapter proteins is required to alter mitochondrial respiration and morphology.

Having determined that engagement of both the MyD88 and TRIF adapter proteins is required to alter mitochondrial respiration, we next sought to validate our pharmacologic results with genetic proof-of-concept using BMDMs isolated from mice lacking either adapter protein. If both MyD88 and TRIF are required to decrease mitochondrial respiration, then loss of either protein should rescue respiration in response to relevant stimuli. Indeed, the loss of either protein was sufficient to increase ATP-linked and maximal respiration in macrophages polarized for 24 h with stimuli that engage both MyD88 and TRIF: Pam3 with Poly I:C (Fig. 2A,B; Appendix Fig. S2A), 50 ng/mL LPS (Fig. 2C), and IMQ with Poly I:C (Appendix Fig. S2B,C).

We then sought to better understand which signaling program downstream of TRIF is required to lower oxidative phosphorylation, hypothesizing that this was an interferon-linked response (Akira and Takeda, 2004). Indeed, treatment of BMDMs with interferon-γ (IFN-γ) along with Pam3, 10 ng/mL LPS, or HKSA for 24 h collapsed mitochondrial respiration (Fig. 2D–F; Appendix Fig. S2D). Furthermore, BMDMs isolated from mice lacking MyD88 treated with IFN-γ along with 10 ng/mL LPS showed a complete rescue in ATP-linked and maximal respiration after 24 h compared to controls (Appendix Fig. S2E). No significant change in respiration was observed when treating macrophages with both Poly I:C and IFN-γ (Fig. 2D), further highlighting the requirement of MyD88. Additionally, macrophages isolated from mice lacking the Type I interferon receptor (IFNAR) had restored respiration in response to co-treatment with Pam3 and IFN-β but not Pam3 and IFN-γ, a Type II interferon that bypasses IFNAR (Fig. 2G,H). Altogether, the data further show that reduced mitochondrial respiration is not a characteristic feature of all pro-inflammatory stimuli, but rather only those that engage both MyD88- and IFN-linked signaling.

Increased glycolysis is another bioenergetic hallmark of pro-inflammatory macrophage activation (Tannahill et al, 2013; Millet et al, 2016; Palsson-Mcdermott et al, 2015; Xie et al, 2016; Semba et al, 2016). To better understand which signaling pathways were required to increase glycolysis, we treated BMDMs with Pam3, 10 ng/mL LPS, Poly I:C, or IFN-γ for 24 h. Only stimuli upstream of MyD88 (Pam3 and 10 ng/mL LPS) substantially increased rates

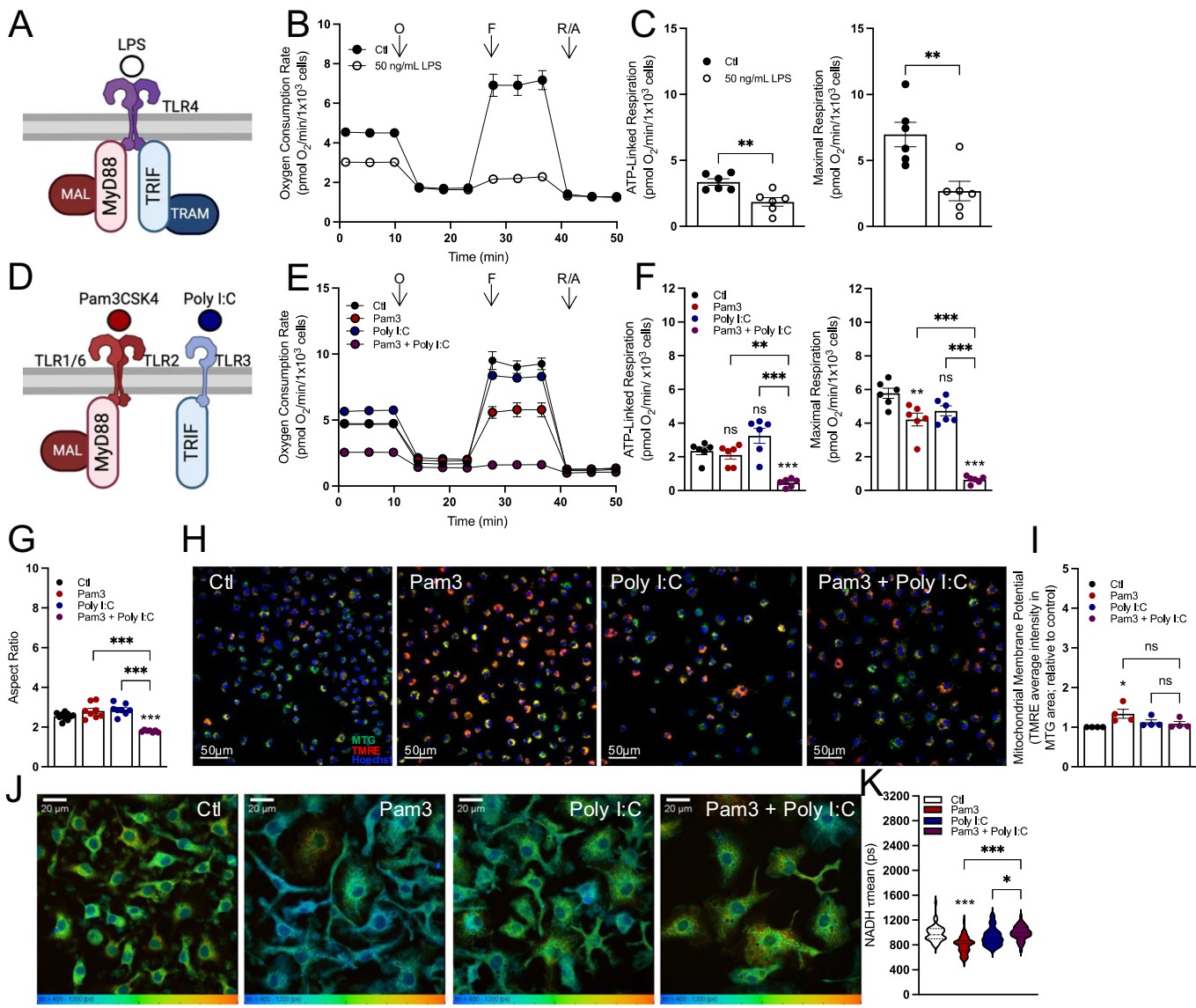

**Figure 1. Various pro-inflammatory stimuli elicit different effects on mitochondrial function after 24 h.**

(A) A schematic of the adapter proteins immediately downstream of TLR4 engaged after lipopolysaccharide (LPS) treatment. (B) Representative oxygen consumption trace for BMDMs treated with 50 ng/mL LPS for 24 h or vehicle control (Ctl). ($n = 1$ biological replicate with five technical replicates). (C) Aggregate ATP-linked and maximal respiration rates for BMDMs treated as in (B) ($n = 6$) (**$P = 0.004$ ATP-linked, **$P = 0.005$ maximal). (D) A schematic of the adapter proteins immediately downstream of either TLR2 or TLR3 engaged after treatment with various pathogen-associated molecular patterns. (E) Representative oxygen consumption trace for BMDMs treated with Pam3 (50 ng/mL), Poly I:C (1 µg/mL), Pam3 + Poly I:C, or vehicle control (Ctl) for 24 h ($n = 1$ biological with five technical replicates). (F) Aggregate ATP-linked and maximal respiration rates for BMDMs treated as in (E) ($n = 6$) (**$P = 0.003$ ATP-linked, ***$P < 0.001$, **$P = 0.005$ maximal). (G) Mitochondrial aspect ratio quantified using FIJI image analysis software for BMDMs treated as in (E) for 24 h. Images are taken from $n = 7$–10 cells from three independent cell preparations (representative images available in Appendix) (***$P < 0.001$). (H) Representative images of BMDMs treated as in (E) for 24 h. Nuclei are stained with Hoechst (1 µg/mL) and mitochondria are stained with MitoTracker Green (MTG; 200 nM) and TMRE (10 nM). (I) Bulk mitochondrial membrane potential as measured by TMRE fluorescence per mitochondrial area detected by MitoTracker Green (MTG) for BMDMs treated as in (E) for 24 h. Data are shown relative to control ($n = 4$) (*$P = 0.03$). (J) Representative images of BMDMs treated as in (E) for 24 h from fluorescence lifetime imaging microscopy (FLIM) analysis of NADH. $\tau_{mean} = 400$–1200 ps. (K) Mean endogenous NADH lifetime ($\tau_{mean}$) for BMDMs as in (J) measured in picoseconds (ps). Each data point represents an individual cell ($n = 50$–80 cells for each condition from a single biological replicate) (***$P < 0.001$, *$P = 0.04$). All data presented in Fig. 1 are mean ± standard error of the mean (SEM) with statistical analysis conducted on biological replicates, each of which included multiple technical replicates, unless otherwise indicated. When not visible, error bars are obscured by the symbol for all oxygen consumption traces. O oligomycin, F FCCP, R/A rotenone/antimycin A. Statistical analysis for (C) was performed as an unpaired, two-tailed, $t$-test. Statistical analysis for (F–I, K) was performed as an ordinary one-way, ANOVA followed by Tukey's post hoc multiple comparisons test. Source data are available online for this figure.

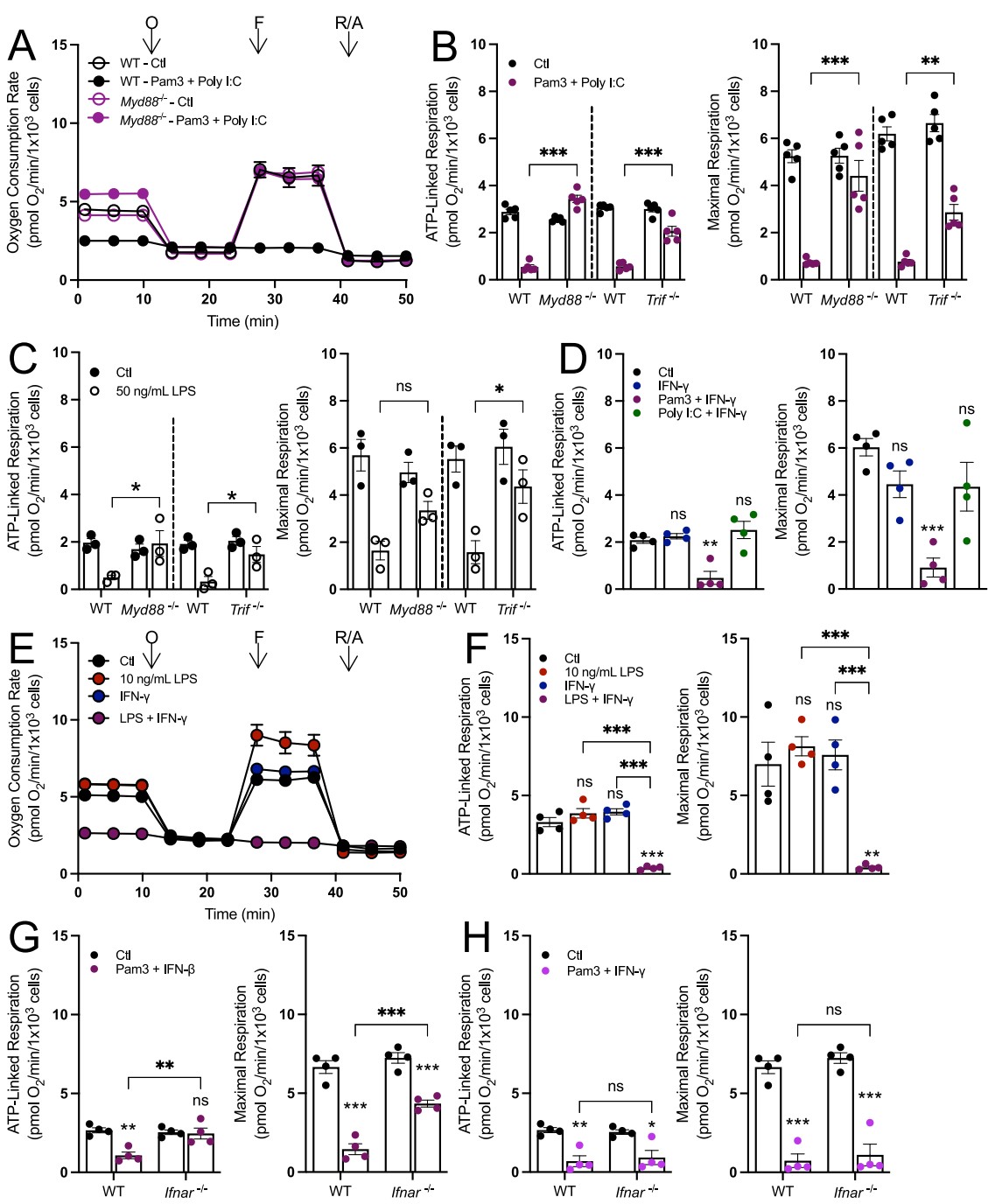

## Accumulation of pro-inflammatory metabolites is not linked to changes in mitochondrial respiration

Having established that not all pro-inflammatory stimuli collapse oxidative phosphorylation, we next aimed to understand precisely

of lactate efflux and intracellular lactate abundance (Appendix Fig. S3A,B). This effect was lost in BMDMs lacking MyD88, establishing the requirement of MyD88-linked signaling for the profound increase in glycolysis (Appendix Fig. S3C). The results further reinforce the lack of a universal bioenergetic phenotype that characterizes all pro-inflammatory stimuli.

which features of the pro-inflammatory response correlated with reduced oxygen consumption. We first applied this approach to study the accumulation of the TCA cycle-linked metabolites succinate and itaconate (Fig. 3A), hypothesizing that engagement of both MyD88 and TRIF would result in enhanced accumulation of these metabolites relative to activating either pathway in isolation. During LPS ± IFN-γ activation, distinct "breaks" occur in the TCA cycle slowing the activity of isocitrate dehydrogenase (IDH) and succinate dehydrogenase (SDH) (Jha et al, 2015; Lampropoulou et al, 2016; Cordes et al, 2016). It is generally accepted that this enzyme inhibition slows oxidative phosphorylation and leads to the accumulation of the metabolites itaconate and

◀  **Figure 2.  Both MyD88- and IFN-linked signaling are required to decrease oxidative phosphorylation.**

(A) Representative oxygen consumption trace for BMDMs harvested from wildtype (WT) or MyD88-null ($Myd88^{-/-}$) mice and treated with either vehicle control (Ctl) or Pam3 (50 ng/mL) + Poly I:C (1 μg/mL) for 24 h ($n = 1$ biological replicate with five technical replicates). (B) Aggregate ATP-linked and maximal respiration rates for BMDMs harvested from WT, $Myd88^{-/-}$, or $Trif^{-/-}$ mice and treated as in (A) ($n = 5$) (***$P < 0.001$, **$P = 0.004$). (C) Aggregate ATP-linked and maximal respiration rates for BMDMs harvested from WT, $Myd88^{-/-}$, or $Trif^{-/-}$ mice and treated with 50 ng/mL LPS for 24 h ($n = 3$) (*$P = 0.04$ ATP-linked $Myd88^{-/-}$, *$P = 0.05$ ATP-linked $Trif^{-/-}$, *$P = 0.02$ Maximal $Trif^{-/-}$). (D) ATP-linked and maximal respiration rates for BMDMs treated with IFN-γ (20 ng/mL), Pam3 + IFN-γ, Poly I:C + IFN-γ, or vehicle control (Ctl) ($n = 4$) (**$P = 0.003$, ***$P < 0.001$). (E) Representative oxygen consumption trace for BMDMs treated with LPS (10 ng/mL), IFN-γ (20 ng/mL), LPS + IFN-γ, or vehicle control (Ctl) for 24 h ($n = 1$ biological replicate with five technical replicates). (F) Aggregate ATP-linked and maximal respiration rates for BMDMs as in (E) ($n = 4$) (***$P < 0.001$, **$P = 0.001$). (G, H) Aggregate ATP-linked and maximal respiration rates for BMDMs harvested from WT and IFNAR-null ($Ifnar^{-/-}$) mice. BMDMs were treated with vehicle control (Ctl) and either (G) Pam3 + IFNβ (20 ng/mL) or (H) Pam3 + IFN-γ ($n = 4$) (G: **$P = 0.002$ ATP-linked WT Ctl vs. WT Pam3 + IFN-β, **$P = 0.007$ ATP-linked WT Pam3 + IFN-β vs. $Ifnar^{-/-}$ Pam3 + IFN-β, ***$P < 0.001$; H: ***$P = 0.004$ ATP-linked WT Ctl vs. WT Pam3 + IFN-γ, *$P = 0.02$ ATP-linked $Ifnar^{-/-}$ Ctl vs. $Ifnar^{-/-}$ Pam3 + IFN-γ, ***$P < 0.001$). All data presented in Fig. 2 are mean ± standard error of the mean (SEM) with statistical analysis conducted on data from biological replicates, each of which included multiple technical replicates, unless otherwise indicated. When not visible, error bars are obscured by the symbol for all oxygen consumption traces. O oligomycin, F FCCP, R/A rotenone/antimycin A. Statistical analysis for (B, C, G, and H) was performed as an ordinary two-way, ANOVA followed by Sídák's post hoc multiple comparisons test. Statistical analysis for (D, F) was performed as an ordinary one-way, ANOVA followed by Tukey's post hoc multiple comparisons test. Source data are available online for this figure.

succinate, both of which can impact cell signaling, post-translational modifications, and the antimicrobial response (Tannahill et al, 2013; Cordes et al, 2015; Ryan et al, 2019; O'Neill and Artyomov, 2019).

Although neither Pam3 nor Poly I:C inhibit oxidative phosphorylation, steady-state levels of intracellular itaconate and succinate still increased substantially after 24-h treatment, and to levels comparable with co-treatment (Fig. 3B). This pattern was reproduced with other pairs of MyD88- and IFN-linked ligands, such as 10 ng/mL LPS with IFN-γ (Fig. 3C) or IMQ with Poly I:C (Appendix Fig. S4A).

Next, we studied BMDMs harvested from $Irg1^{-/-}$ mice to further understand the extent to which accumulation of succinate and itaconate is linked with the collapse in respiration from LPS ± IFN-γ. Siphoning carbon out of the TCA cycle to generate itaconate, and the subsequent inhibition of SDH, is thought to contribute to the restricted oxidative phosphorylation observed upon classical macrophage activation (Lampropoulou et al, 2016). However, the loss of $Irg1$ and the inability to accumulate itaconate and succinate had no effect on the respiratory inhibition observed in response to 24-h treatment with either LPS and IFN-γ (Fig. 3D–F) or Pam3 and Poly I:C (Appendix Figs. S4B,C). Additionally, as has been previously reported (Vijayan et al, 2019), human monocyte-derived macrophages (HMDMs) do not exhibit respiratory inhibition in response to stimulation with LPS (Fig. 3G). However, HMDMs readily accumulate itaconate and succinate (Fig. 3H). Altogether, the data show that macrophages do not need to repurpose mitochondria away from oxidative phosphorylation to directly support the generation of pro-inflammatory metabolites.

As Fig. 3 shows, pro-inflammatory macrophages can accumulate signaling metabolites while maintaining oxidative phosphorylation, we sought to better understand whether the TCA cycle remained fully functional. After 24 h, citrate abundance sharply increased only upon co-treatment with Pam3 and Poly I:C and not with either ligand alone (Fig. 4A), confirming a "break" in the TCA cycle as NO-mediated inhibition of IDH and aconitase activity occurs only with activation of both MyD88 and IFN signaling (Bailey et al, 2019). To better examine individual enzymes and pathways, we measured oxygen consumption in plasma membrane-permeabilized BMDMs to directly provide mitochondria with specific respiratory substrates (Appendix Fig. S5A). BMDMs were treated with Pam3, Poly I:C, or both ligands, and offered various substrate pairs to better understand enzymatic capacity in response

to these stimuli. We observed the same pattern regardless of whether permeabilized cells were offered pyruvate with malate, glutamate with malate, succinate with rotenone, or citrate: only co-treatment with Pam3 and Poly I:C reduced oxygen consumption rates (Fig. 4B). The results, particularly with citrate-driven respiration, further indicate that both a MyD88 and IFN signal are required for respiratory inhibition and reduced IDH activity. As treatment with Pam3 or Poly I:C alone can accumulate the SDH inhibitor itaconate (Fig. 3B), it was unexpected that succinate-driven respiration was unchanged. However, a time-course revealed this is likely due to exogenously added succinate outcompeting endogenous itaconate (Appendix Fig. S5B,C), which is well established as a non-covalent inhibitor of SDH (Cordes et al, 2016). Altogether, these results suggest that either Pam3 or Poly I:C alone do not elicit 'breaks' in the TCA cycle after 24 h, and engaging both MyD88- and IFN-linked pathways is required.

We then hypothesized that Pam3- or Poly I:C-treated BMDMs accumulate these signaling metabolites not by slowing metabolite consumption but rather by increasing synthesis. To test this, we conducted stable isotope tracing in BMDMs with $^{13}C_6$-glucose or $^{13}C_5$-glutamine. BMDMs were pretreated with TLR agonists for 18 h, after which cells were washed and given new medium containing $^{13}C_6$-glucose or $^{13}C_5$-glutamine for 6 h. Tracing with isotopically labeled glucose or glutamine resulted in distinct labeling patterns, with glucose enriching the M + 2 isotopologues of TCA cycle metabolites (M + 1 for itaconate) and glutamine enriching the M + 4 isotopologues (M + 5 for α-ketoglutarate) (Fig. 4C). BMDMs treated with Pam3 or Poly I:C showed enrichment from glucose into α-ketoglutarate and fumarate, the two metabolites immediately after the breaks in the TCA cycle, while co-treatment with both ligands displayed minimal enrichment into either metabolite (Fig. 4D). Moreover, Pam3-treated BMDMs showed increased enrichment from glucose into several TCA cycle metabolites, while this was predictably decreased in BMDMs co-treated with both Pam3 and Poly I:C in a manner consistent with reduced IDH activity (Fig. 4E; full isotopologue distributions from both $^{13}C_6$-glucose and $^{13}C_5$-glutamine can be found in Appendix Table S1). Overall, the respirometry and stable isotope tracing data suggest macrophages treated with either Pam3 or Poly I:C for 24 h maintain relative TCA cycle flux and accumulate signaling metabolites by enhancing metabolite synthesis rather than decreasing consumption.

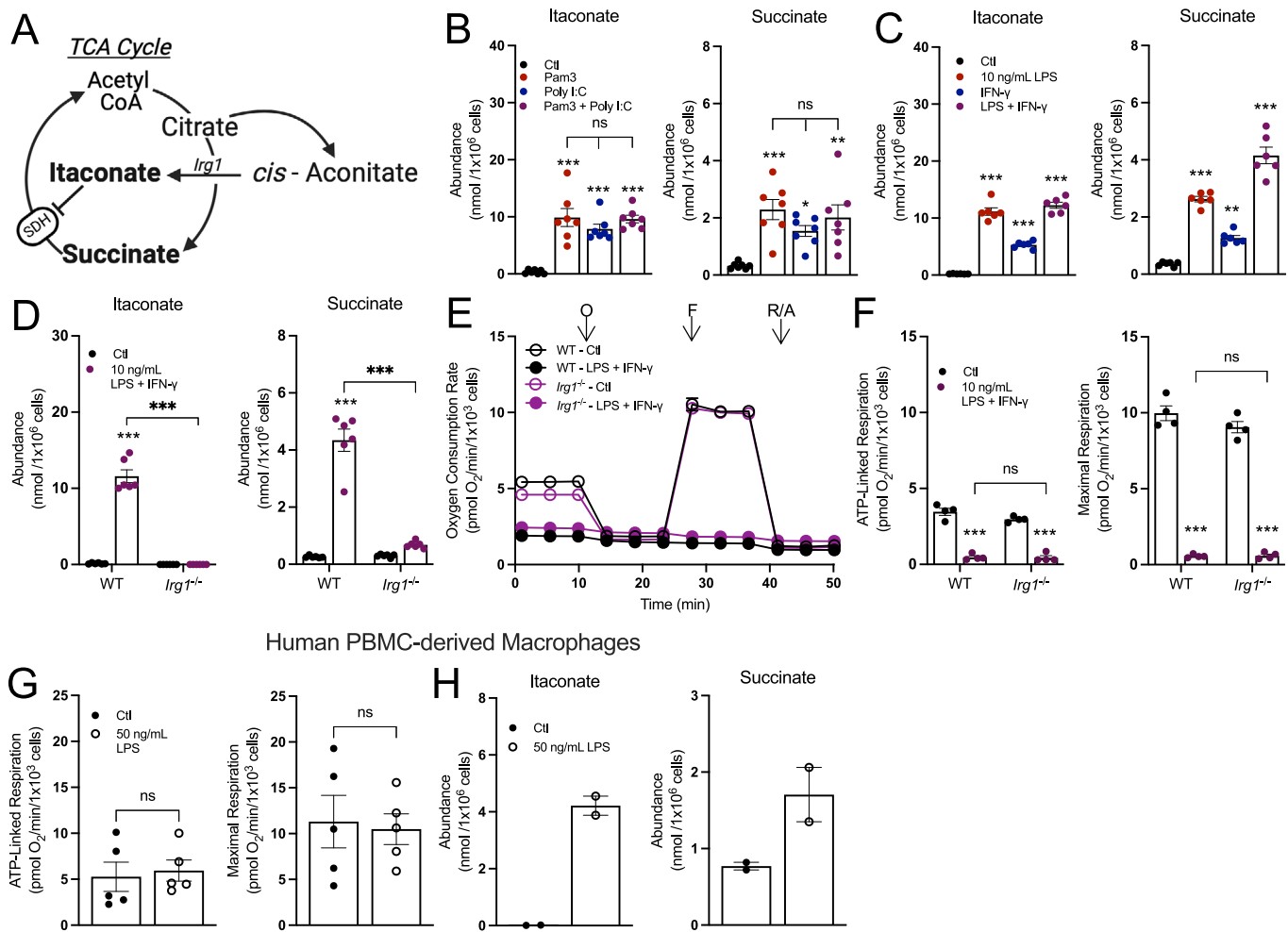

**Figure 3. Accumulation of succinate and itaconate is independent of reductions in oxidative phosphorylation.**

(A) A schematic depicting itaconate and succinate in the context of the TCA cycle. (B) Intracellular levels of itaconate and succinate from BMDMs treated with vehicle control (Ctl), Pam3 (50 ng/mL), Poly I:C (1 µg/mL), or Pam3 + Poly I:C for 24 h ($n = 7$) (***$P < 0.001$, *$P = 0.04$, **$P = 0.003$). (C) Intracellular levels of itaconate and succinate from BMDMs treated with vehicle control, LPS (10 ng/mL), IFN-γ (20 ng/mL), or LPS + IFN-γ for 24 h ($n = 6$) (***$P < 0.001$, **$P = 0.003$). (D) Intracellular levels of itaconate and succinate in BMDMs harvested from wildtype (WT) or *cis*-aconitate decarboxylase (CAD)-null (*Irg1*$^{-/-}$) mice and treated with vehicle control (Ctl) or LPS + IFN-γ for 24 h ($n = 6$) (***$P < 0.001$). (E) Representative oxygen consumption trace for BMDMs harvested from WT or *Irg1*$^{-/-}$ mice and treated with either vehicle control (Ctl) or LPS + IFN-γ for 24 h ($n = 1$ biological replicate with 5 technical replicates). (F) Aggregate ATP-linked and maximal respiration rates for BMDMs as in (E) ($n = 4$) (***$P < 0.001$). (G) ATP-linked and maximal respiration rates in human monocyte-derived macrophages (HMDMs) treated with vehicle control (Ctl) or 50 ng/mL LPS for 24 h ($n = 5$). (H) Intracellular levels of itaconate and succinate from HMDMs treated as in (G). All data presented in Fig. 3 for BMDMs are mean ± standard error of the mean (SEM) with statistical analysis conducted on data from biological replicates, each of which included multiple technical replicates, unless otherwise indicated. Data in (G, H) are mean ± spread from $n = 2$ individual human donors. When not visible, error bars are obscured by the symbol for all oxygen consumption traces. O oligomycin. F FCCP, R/A rotenone/antimycin A. Statistical analysis for (B, C) was performed as an ordinary one-way, ANOVA followed by Tukey's post hoc multiple comparisons test. Statistical analysis for (D, F) was performed as an ordinary two-way, ANOVA followed by Šídák's post hoc multiple comparisons test. Statistical analysis for (G) was performed as an unpaired, two-tailed *t*-test. Source data are available online for this figure.

## Respiratory inhibition does not enhance pro-inflammatory macrophage activation

After determining that the accumulation of itaconate and succinate is independent of decreased mitochondrial respiration, we next asked what other features of the macrophage pro-inflammatory response might be augmented by respiratory inhibition. Indeed, a priming effect from IFN-γ on LPS-induced *Nos2* gene expression and nitric oxide (NO) production has been appreciated for decades (Lowenstein et al, 1993; Lorsbach et al,

1993). We, therefore, hypothesized that combinations of MyD88- and IFN-linked ligands could amplify pro-inflammatory gene expression at 24 h. As expected, Pam3 and Poly I:C synergistically increased the expression of many pro-inflammatory genes (Fig. 5A; Appendix Fig. S6A) after 24 h, as did IMQ with Poly I:C (Appendix Fig. S6B) and 10 ng/mL LPS with IFN-γ (Appendix Fig. S6C). Co-treatment with Pam3 and Poly I:C also increased IL-1β and IL-12 release as well as nitrite production (Fig. 5B). As these co-treatments also decreased mitochondrial respiration (Fig. 1), the results establish an association between enhanced

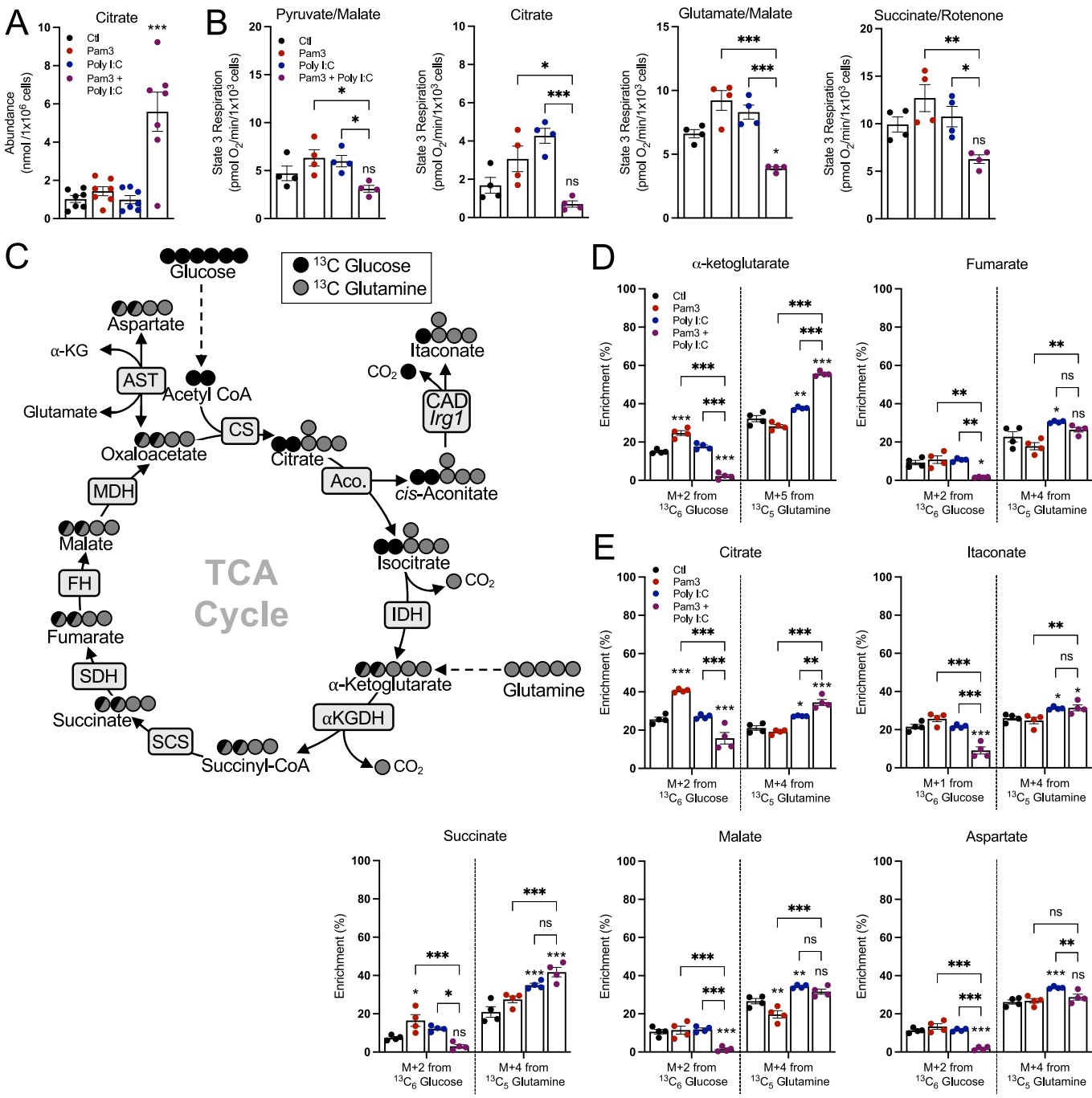

pro-inflammatory gene expression and decreased mitochondrial respiration after 24 h.

To determine if this relationship is causative, we analyzed respiration and pro-inflammatory markers in BMDMs isolated from mice lacking inducible nitric oxide synthase (*Nos2*, iNOS). NO production via *Nos2* is the dominant mechanism by which oxidative phosphorylation decreases in macrophages (Clementi et al, 1998; Palmieri et al, 2020a; Seim et al, 2023; Bailey et al, 2019; Van den Bossche et al, 2016). As predicted, respiratory inhibition from co-treatment with Pam3 and Poly I:C was almost entirely rescued upon loss of iNOS (Fig. 5C; Appendix Fig. S7A,B).

However, in line with previous reports, the expression of many pro-inflammatory genes and release of cytokines was not significantly reduced (Fig. 5D,E; Appendix Fig. S7C) (Bailey et al, 2019; Palmieri et al, 2020a). The results reinforce that prevention of the NO-mediated inhibition of the mitochondrial respiratory chain and TCA cycle does not affect pro-inflammatory gene expression.

To further understand the relationship between respiratory chain activity and classical macrophage activation, we measured whether inhibition of oxidative phosphorylation could enhance gene expression (Fig. 5F). When used in conjunction with a MyD88- or TRIF-linked TLR agonist for 24 h, blocking complex I

**Figure 4. Not all pro-inflammatory stimuli induce characteristic "breaks" to the TCA cycle.**

(A) Intracellular levels of citrate from BMDMs treated with vehicle control (Ctl), Pam3 (50 ng/mL), Poly I:C (1 μg/mL), or Pam3 + Poly I:C for 24 h ($n = 7$) (***$P < 0.001$).
(B) State 3 respiration in plasma membrane-permeabilized BMDMs treated as in (A). Permeabilized BMDMs were offered pyruvate/malate, citrate, glutamate/malate, or succinate/rotenone as respiratory substrates ($n = 4$) (Pyr/Mal: *$P = 0.02$ Pam3 vs. Pam3 + Poly I:C, *$P = 0.04$ Poly I:C vs. Pam3 + Poly I:C; Cit: *$P = 0.01$ Pam3 vs Pam3 + Poly I:C, ***$P < 0.001$; Glu/Mal: ***$P < 0.001$, *$P = 0.01$ Ctl vs. Pam3 + Poly I:C; Succ/Rot: **$P = 0.003$ Pam3 vs. Pam3 + Poly I:C, *$P = 0.04$ Poly I:C vs. Pam3 + Poly I:C). (C) Schematic depicting the labeling patterns of TCA cycle metabolites from uniformly labeled $^{13}C_6$-glucose (black circles) or $^{13}C_5$-glutamine (gray circles). CS citrate synthase, Aco. aconitase, IDH isocitrate dehydrogenase, α-KGDH alpha-ketoglutarate dehydrogenase, SCS succinyl CoA synthetase, SDH succinate dehydrogenase, FH fumarate hydratase, MDH malate dehydrogenase, AST aspartate transaminase. (D, E) Metabolite isotopologues from BMDMs offered either $^{13}C_6$-glucose or $^{13}C_5$-glutamine and treated as in (A). BMDMs were treated with stimuli in medium containing unlabeled substrates for 16 h, then changed to medium containing $^{13}$C-labeled substrates along with relevant stimuli for 6 h ($n = 4$) (D: αKG: ***$P < 0.001$, **$P = 0.01$; Fumarate ($^{13}C_6$-glucose):**$P = 0.004$ Pam3 vs. Pam3 + Poly I:C, **$P = 0.005$ Poly I:C vs. Pam3 + Poly I:C, *$P = 0.02$ Ctl vs. Pam3 + Poly I:C, Fumarate ($^{13}C_5$-glutamine): **$P = 0.008$ Pam3 vs. Pam3 + Poly I:C, *$P = 0.02$ Ctl vs Poly I:C; E: Citrate: ***$P < 0.001$, **$P = 0.006$, *$P = 0.02$; Itaconate: ***$P < 0.001$, **$P = 0.008$, *$P = 0.05$, *$P = 0.04$; Succ: *$P = 0.01$ Ctl vs. Pam3, *$P = 0.01$ Poly I:C vs. Pam3 + Poly I:C; Malate: ***$P < 0.001$, **$P = 0.007$ Ctl vs Pam3, **$P = 0.002$ Ctl vs. Poly I:C; Aspartate: ***$P < 0.001$, **$P = 0.009$). All data presented in Fig. 4 are mean ± standard error of the mean (SEM) with statistical analysis conducted on data from biological replicates, each of which included multiple technical replicates, unless otherwise indicated. Statistical analysis for (A, B) was performed as an ordinary one-way, ANOVA followed by Tukey's post hoc multiple comparisons test. Statistical analysis for (D, E) was performed as an ordinary two-way, ANOVA followed by Tukey's post hoc multiple comparisons test. Source data are available online for this figure.

activity (with piericidin A), complex III activity (with antimycin A), adenine nucleotide transport (with carboxyatractyloside), complex V activity (with oligomycin), or mitochondrial membrane potential maintenance (with Bam15 and oligomycin) did not enhance pro-inflammatory gene expression or cytokine release above that of Pam3, Poly I:C, or 10 ng/mL LPS alone (Fig. 5G–I; Appendix Fig. S8A–C). Additionally, the mitochondrial effector compounds had no effect on phagocytosis in combination with Pam3, and some even induced a moderate defect in Poly I:C driven phagocytosis (Fig. 5J). Importantly, the inhibitors were used at the lowest concentration that elicited a maximal effect on respiration, and none of the compounds decreased cell counts relative to TLR agonism or affected the pro-inflammatory response in the absence of a TLR agonist (Appendix Fig. S8D–F).

Finally, we CRISPR-edited immortalized primary macrophages to lack *Ndufs4*, a nuclear-encoded subunit of mitochondrial complex I. Despite a profound reduction in NADH oxidation that manifested in reduced ATP-linked and maximal respiration in macrophages (Fig. 5K–N), we observed no change in Pam3-induced pro-inflammatory gene expression in *Ndufs4*-depleted macrophages (Fig. 5O). Together, the pharmacologic and genetic loss-of-function studies indicate that the relationship between macrophage pro-inflammatory gene expression and mitochondrial respiration at 24 h is largely associative rather than causative.

## The induction of the pro-inflammatory response does not temporally align with respiratory inhibition

Although we observed no causal relationship between respiratory inhibition and macrophage activation at 24 h, this did not obviate a role for oxidative phosphorylation at an earlier timepoint and during the induction of the pro-inflammatory response. Upon TLR agonism, the expression of many genes associated with the pro-inflammatory response peaks within minutes or hours and recedes soon after to taper inflammation (Cheng et al, 2021; Bhatt et al, 2012; Lauterbach et al, 2019). We found that across eleven different pro-inflammatory genes, expression increased sharply soon after treatment with 50 ng/mL LPS, peaking within the first 3–6 h (Fig. 6A; Appendix Fig. S9A). However, much of our understanding regarding the role of oxidative phosphorylation in the pro-inflammatory response is based on data in response to LPS treatment ± IFN-γ for 24 h or longer (Bailey et al, 2019; Mills et al,

2016; Van den Bossche et al, 2016; Lampropoulou et al, 2016; Jha et al, 2015; Seim et al, 2019). We, therefore, sought to understand whether mitochondrial energetics were altered within a timeframe commensurate with the peak induction of canonical pro-inflammatory genes.

We measured rates of oxygen consumption at timepoints between 1 and 12 h after treatment with 50 ng/mL LPS and observed no significant defect in maximal respiration until 6 h and later (Fig. 6B). These initial findings suggest that even at earlier timepoints, impaired oxidative phosphorylation does not regulate the induction of pro-inflammatory gene expression. As nitric oxide mediates the inhibition of mitochondrial respiration, we also measured nitrite levels (a stable product of nitric oxide degradation) at each timepoint (Fig. 6C). Indeed, mitochondrial respiratory chain inhibition was more closely aligned with the timeframe of nitrite accumulation rather than gene expression.

We then measured other features of mitochondrial function during an early timeframe, specifically examining whether co-treatment with Pam3 and Poly I:C displays the same profound mitochondrial alterations at 4 h (when gene expression is closer to its peak) as at 24 h. However, we observed no alterations to mitochondrial respiration with either ligand alone or in combination after 4 h (Fig. 6D,E). Moreover, we also observed no change in membrane potential for all groups relative to vehicle controls (Fig. 6F,G). We did, however, observe an increase in NADH fluorescence lifetime for all treatment groups relative to vehicle controls, suggesting that changes to the NADH pool size or redox status can be altered during macrophage activation while maintaining rates of oxidative phosphorylation (Fig. 6H,I).

We then measured whether steady-state metabolite levels could be increased after only 4-h treatment and prior to reductions in oxygen consumption. Indeed, itaconate and succinate levels increased at this relatively early timepoint for each treatment group, further suggesting that the accumulation of these signaling metabolites is independent of respiratory chain inhibition (Fig. 6J). We also observed no defects in permeabilized cell respirometry after 4-h treatment (Fig. 6K). Furthermore, unlike the 24-h timepoint (Fig. 4D), BMDMs co-treated with Pam3 and Poly I:C did not have decreased incorporation of $^{13}C_6$-glucose into either α-ketoglutarate or fumarate, the two metabolites immediately downstream of the canonical 'breaks' in the TCA cycle (Fig. 6L).

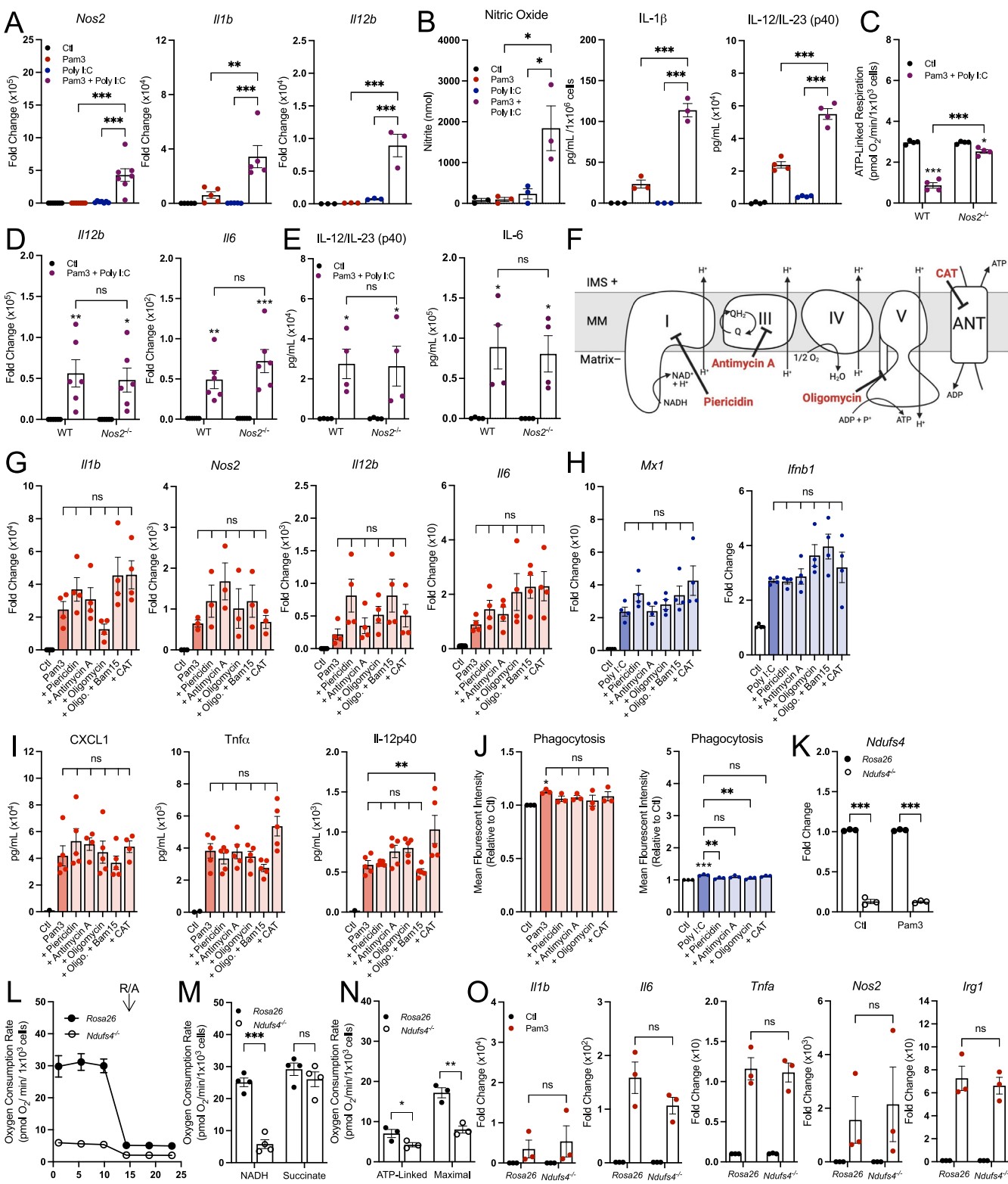

◀ **Figure 5. Respiratory inhibition does not enhance pro-inflammatory macrophage activation at 24 h.**

(A) Pro-inflammatory gene expression in BMDMs treated with vehicle control (Ctl), Pam3 (50 ng/mL), Poly I:C (1 µg/mL), or Pam3 + Poly I:C for 24 h ($n = 3–7$) (***$P < 0.001$, **$P = 0.001$). (B) Levels of nitrite ($n = 3$), IL-1β ($n = 3$), and IL-12/IL-23 (p40) ($n = 4$) in medium collected from BMDMs treated as in (A) (*$P = 0.01$ Pam3 vs. Pam3 + Poly I:C, *$P = 0.02$ Poly I:C vs. Pam3 + Poly I:C, ***$P < 0.001$). (C) ATP-linked respiration rates for BMDMs harvested from wildtype (WT) or iNOS-null ($Nos2^{-/-}$) treated with vehicle control (Ctl) or Pam3 + Poly I:C for 24 h ($n = 4$) (***$P < 0.001$, *$P = 0.02$). (D, E) Pro-inflammatory gene expression (D) ($n = 6$) and secreted cytokine levels (E) ($n = 4$) for IL-6 and IL-12β from BMDMs treated as in (C) (D: $Il12b$: **$P = 0.009$,*$P = 0.03$; $Il6$: **$P = 0.007$, ***$P < 0.001$; E: IL-12/IL-23 (p40): *$P = 0.04$ WT Ctl vs. Pam3 + Poly I:C, *$P = 0.05$ $Nos2^{-/-}$ Ctl vs. Pam3 + Poly I:C; IL-6: *$P = 0.02$ WT Ctl vs. Pam3 + Poly I:C, *$P = 0.03$ $Nos2^{-/-}$ Ctl vs. Pam3 + Poly I:C). (F) Schematic of mitochondrial respiratory chain inhibitors used in (G–J). CAT carboxyatractyloside. (G, H) Pro-inflammatory gene expression for BMDMs treated with vehicle controls (Ctl) and either Pam3 (G) ($n = 3–4$) or Poly I:C (H) ($n = 4$) along with the following mitochondrial effector compounds for 24 h: piericidin (100 nM), antimycin A (30 nM), oligomycin (10 nM), oligomycin (10 nM) + Bam15 (3 µM), or CAT (30 µM). (I) Cytokine levels from the spent medium of BMDMs treated as in (G) ($n = 5$) (**$P = 0.007$). (J) Phagocytosis of heat-killed, fluorescently pre-labeled $E.$ $coli$ particles in BMDMs treated as in (G, H) ($n = 3$) (*$P = 0.05$, ***$P < 0.001$, **$P = 0.005$ Poly I:C vs. Poly I:C + Piericidin, **$P = 0.004$ Poly I:C vs. Poly I:C + Oligomycin). (K) Gene expression after CRISPR-mediated knockdown of $Ndufs4$ in immortalized BMDMs (iBMDMs) relative to control guides targeting $Rosa26$ ($n = 3$) (***$P < 0.001$). (L, M) Representative oxygen consumption trace (L) and collated biological replicates (M) for iBMDMs as in (K). Cells were 'double permeabilized' with rPFO (3 nM) and alamethicin (3 ng/mL) to offer NADH or succinate directly to the respiratory chain. R/A, rotenone/antimycin A [$n = 1$ biological with 5 technical replicates for (L); $n = 4$ for (M)]. Where not visible, error bars are obscured by the symbol (***$P < 0.001$). (N) ATP-linked and maximal respiration rates in intact iBMDMs offered glucose, pyruvate, and glutamine as respiratory substrates ($n = 4$) (*$P = 0.04$, **$P = 0.005$). (O) Pro-inflammatory gene expression in CRISPR-edited iBMDMs treated with vehicle control (Ctl) or Pam3 (50 ng/mL) for 24 h ($n = 4$). All data presented in Fig. 5 are mean ± standard error of the mean (SEM) with statistical analysis conducted on data from biological replicates, each of which included multiple technical replicates, unless otherwise indicated. Cytokine levels in (B, E, I) below the threshold of detection are given as zero. Statistical analysis for (A, B, G–J) was performed as an ordinary one-way, ANOVA followed by Tukey's post hoc multiple comparisons test. Statistical analysis for (C, K, M, O) was performed as an ordinary two-way, ANOVA followed by Sídák's post hoc multiple comparisons test. Statistical analysis for (D, E) was performed as an ordinary two-way, ANOVA followed by Tukey's post hoc multiple comparisons test. Statistical analysis for (N) was performed as an unpaired, two-tailed $t$-test. Source data are available online for this figure.

Lastly, we examined whether pharmacologic inhibition of oxidative phosphorylation could increase gene expression after 4-h treatment with Pam3, Poly I:C, or LPS. Consistent with other data, we observed no enhancement of pro-inflammatory gene expression (Fig. 6M,N; Appendix Fig. S9B–D). In total, our data suggest reductions in mitochondrial respiration are not essential for the induction of pro-inflammatory gene expression in BMDMs.

## Peritoneal macrophages can be activated in vivo without decreasing mitochondrial respiration

Finally, we assessed the effect of in vivo macrophage activation on mitochondrial energetics. Macrophages were intraperitoneally activated with Pam3 or LPS for 24 h and assessed ex vivo for 3 h after harvest (Fig. 7A; Appendix Fig. S10). Both treatments increased IL-6 in the serum and IL-6 and IL-12 in the lavage fluid (Fig. 7B,C; Appendix Fig. S10A,B), and LPS treatment resulted in increased pro-inflammatory gene expression (Fig. 7D). In ex vivo peritoneal macrophages, lactate efflux and intracellular lactate accumulation increased upon activation, again linking glycolysis with classical macrophage activation (Fig. 7E,F; Appendix Fig. S10C,D). At the same time, however, oxygen consumption rates were unchanged (Fig. 7G,H; Appendix Fig. S10E,F), further separating mitochondrial energetics from the induction of a pro-inflammatory response. Additionally, ex vivo peritoneal macrophages increased intracellular accumulation of itaconate and succinate but neither citrate nor α-ketoglutarate, showing again that pro-inflammatory metabolites can accumulate without conventional "breaks" in the TCA cycle (Fig. 7I; Appendix Fig. S10G). To validate the model system, we activated peritoneal macrophages for 24 h in vitro and observed respiratory inhibition in response to LPS but not Pam3, showing that peritoneal macrophages behave similarly to BMDMs in vitro but do not reproduce this phenotype when activated in vivo (Appendix Fig. S10H). Altogether, our work provides several independent lines of evidence that reductions in oxidative phosphorylation are not essential for the macrophage pro-inflammatory response.

## Discussion

Macrophage activation is often bifurcated as classical (Type I) pro-inflammatory activation or alternative (Type II) anti-inflammatory activation (Murray, 2017), and in vitro studies of these activation states have shown these opposing functions are associated with equally polarized metabolic phenotypes (Van den Bossche et al, 2017). However, it is increasingly appreciated that macrophage polarization exists across a broad spectrum of activation states (Martinez and Gordon, 2014). As such, it is reasonable that the metabolic phenotypes accompanying effector function also reside on a broad, flexible continuum (Divakaruni et al, 2018b; Wang et al, 2018; Seim et al, 2019).

Our results highlight that mitochondrial remodeling during macrophage activation is signal-specific and are in line with previous reports showing TLR agonism induces signal-specific reprogramming of the macrophage lipidome (Hsieh et al, 2020). While both MyD88- and TRIF-dependent signaling pathways are required to alter mitochondrial respiration, the data suggest upregulation of glycolysis is largely driven by MyD88-dependent signaling. Additionally, our data indicate macrophages can accumulate succinate and itaconate without collapsing oxidative phosphorylation and "breaking" the TCA cycle, but rather by rerouting intermediary metabolism to support both oxidative phosphorylation and synthesis of signaling metabolites. 24-h treatment with either Pam3 or Poly I:C resulted in different TCA cycle enrichment patterns from glucose or glutamine despite similar rates of oxidative phosphorylation, suggesting signal-specific regulation of oxidative mitochondrial metabolism.

The inability to amplify pro-inflammatory gene expression with respiratory chain inhibitors or CRISPR-mediated loss of $Ndufs4$ in immortalized macrophages could add clarity to existing observations. For example, myeloid-specific loss of $Ndufs4$ in mice can result in an improved pro-inflammatory response, whereas no change was observed in response to myeloid-specific $Mpc1$ loss (Cai et al, 2023; Ran et al, 2023). Although these results could point to a critical role in localized inhibition of the respiratory chain and

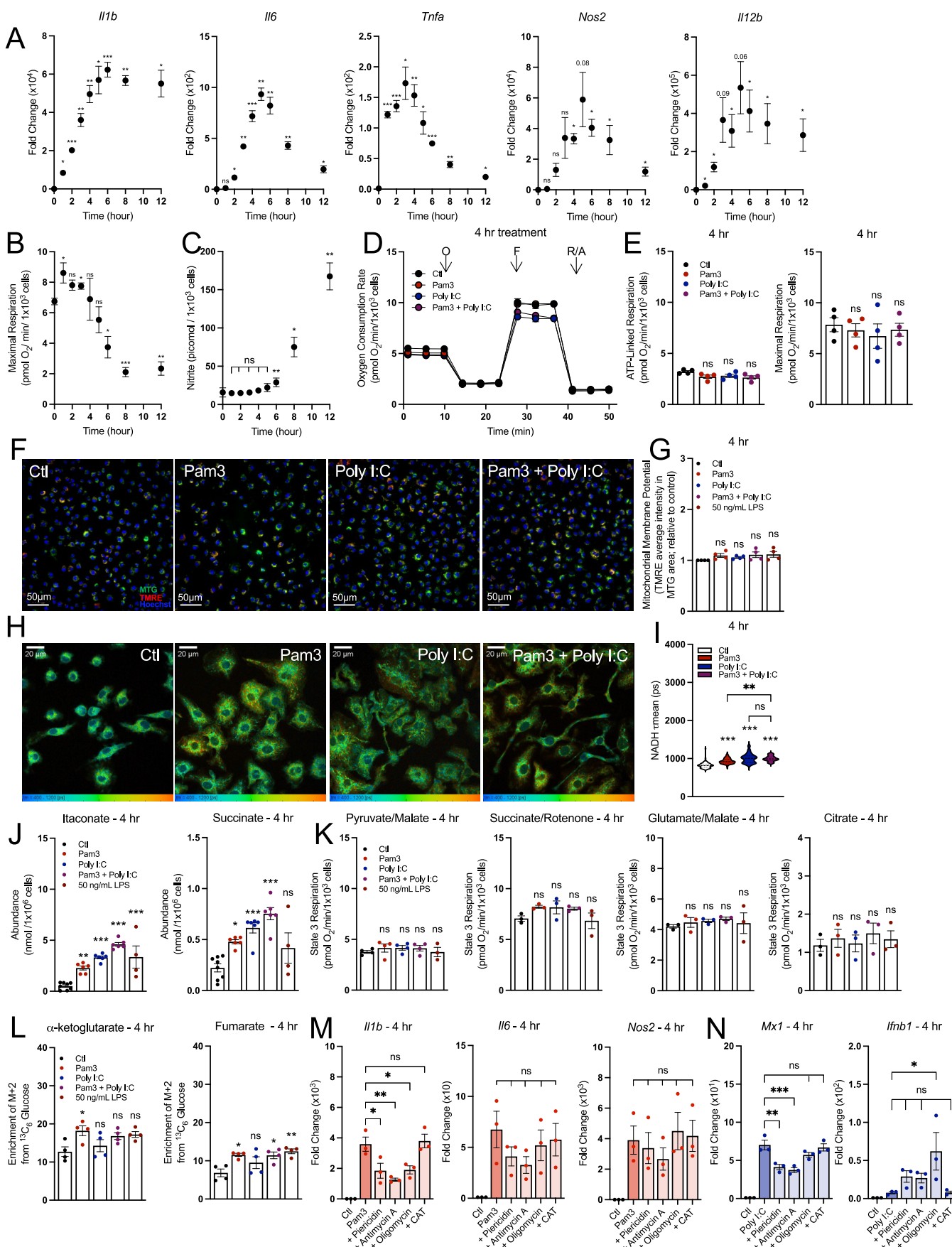

Figure 6. Induction of the pro-inflammatory response does not temporally align with respiratory inhibition.

(A) Pro-inflammatory gene expression in BMDMs treated with vehicle control (Ctl) or 50 ng/mL LPS harvested at multiple timepoints ($n = 3$–4) (*Il1b*: *$P = 0.01$ 1 h, ***$P < 0.001$ 2 h, **$P = 0.009$ 3 h, **$P = 0.002$ 4 h, *$P = 0.02$ 5 h, ***$P < 0.001$ 6 h, **$P = 0.002$ 8 h, *$P = 0.02$ 12 h; *Il6*: *$P = 0.02$ 2 h, **$P = 0.004$ 3 h, ***$P < 0.001$ 4 h, **$P = 0.004$ 5 h, **$P = 0.002$ 6 h, **$P = 0.001$ 8 h, *$P = 0.01$ 12 h; *Tnfa*: ***$P < 0.001$ 1, 2, 6 h, *$P = 0.02$ 3 h, **$P = 0.003$ 4 h, *$P = 0.03$ 5 h, **$P = 0.004$ 8 h, *$P = 0.02$ 12 h; *Nos2*: *$P = 0.01$ 4 h, *$P = 0.02$ 6 h, *$P = 0.04$ 8 h, *$P = 0.03$ 12 h; *Il12b*: *$P = 0.02$ 1, 2 h, *$P = 0.04$ 4 h, *$P = 0.03$ 6 h, *$P = 0.05$ 8 h, *$P = 0.04$ 12 h). (B) Maximal respiration rates for BMDMs as in (A) ($n = 3$–4) (*$P = 0.03$ 1 h, *$P = 0.05$ 3 h, *$P = 0.03$ 6 h, ***$P < 0.001$ 8 h, **$P = 0.004$ 12 h). (C) Levels of nitrite in medium collected from BMDMs treated as in (A) ($n = 3$–4; values below the level of detection are represented as zero) (**$P = 0.007$ 6 h, *$P = 0.02$ 8 h, **$P = 0.001$ 12 h). (D) Representative oxygen consumption trace for BMDMs treated with Pam3 (50 ng/mL), Poly I:C (1 µg/mL), Pam3 + Poly I:C, or vehicle control (Ctl) for 4 h ($n = 1$ biological with 5 technical replicates). (E) Aggregate ATP-linked and maximal respiration rates for BMDMs treated as in (D) ($n = 4$). (F) Representative images of BMDMs treated as in (D) for 4 h. Nuclei are stained with Hoechst (1 µg/mL) and mitochondria are stained with MitoTracker Green (MTG; 200 nM) and TMRE (10 nM). (G) Bulk mitochondrial membrane potential as measured by TMRE fluorescence per mitochondrial area detected by MitoTracker Green (MTG) for BMDMs treated as in (D) for 4 h. Data are shown relative to control ($n = 4$). (H) Representative images of BMDMs treated as in (D) for 4 h. from FLIM analysis of NADH. $\tau_{mean} = 400$–1200 ps. (I) Mean endogenous NADH lifetime ($\tau_{mean}$) for BMDMs as in (H) measured in picoseconds (ps). Each data point represents an individual cell ($n = 87$–135 cells for each condition from a single biological replicate) (***$P < 0.001$, **$P = 0.001$). (J) Intracellular abundances of itaconate and succinate in BMDMs treated as in (D) for 4 h ($n = 4$) (**$P = 0.006$, ***$P < 0.001$, *$P = 0.03$). (K) State 3 respiration from plasma membrane-permeabilized BMDMs treated as in (D) for 4 h. Permeabilized BMDMs were offered pyruvate/malate, citrate, glutamate/malate, or succinate/rotenone as respiratory substrates ($n = 4$). (L) M + 2 isotopologues of α-ketoglutarate and fumarate from $^{13}C_6$-glucose tracing in BMDMs treated as in (D) for 4 h ($n = 4$) (αKG: *$P = 0.04$; fumarate: *$P = 0.03$ Ctl vs. Pam3 and Ctl vs Pam3 + Poly I:C, **$P = 0.007$). (M, N) Pro-inflammatory gene expression for BMDMs treated with vehicle controls (Ctl) and either Pam3 (M) ($n = 3$–4) or Poly I:C (N) ($n = 4$) along with the following mitochondrial effector compounds for 24 h: piericidin (100 nM), antimycin A (30 nM), oligomycin (10 nM), or CAT (30 µM CAT). ($n = 3$) (M: *$P = 0.04$ Pam3 vs Pam3 + Piericidin, **$P = 0.005$, *$P = 0.05$ Pam3 vs. Pam3 + Oligomycin; N: **$P = 0.002$, ***$P < 0.001$, *$P = 0.04$). All data presented in Fig. 6 are mean ± standard error of the mean (SEM) with statistical analysis conducted on data from biological replicates, each of which included multiple technical replicates, unless otherwise indicated. When not visible, error bars are obscured by the symbol for all oxygen consumption traces. O oligomycin, F FCCP, R/A rotenone/antimycin A. Statistical analysis for (A–C) was performed as a paired, two-tailed *t*-test. Statistical analysis for (E, G, I, J–N) was performed as an ordinary one-way, ANOVA followed by Tukey's post hoc multiple comparisons test. Source data are available online for this figure.

oxidative phosphorylation, our acute pharmacologic and genetic studies show electron transport chain inhibition does not augment the pro-inflammatory response. As such, further work may be required to understand the developmental consequences of losing complex I activity and respiratory chain function during myeloid cell development.

This work also highlights the importance of considering the duration, concentration, and type of stimuli used to elicit an in vitro pro-inflammatory response. Many of the hallmark mitochondrial alterations only occur under conditions of high nitric oxide production triggered by a combined MyD88 and IFN response. We and others have shown that human monocyte-derived macrophages that accumulate much smaller concentrations of nitric oxide do not display this characteristic mitochondrial repurposing (Palmieri et al, 2020b). Thus, some mitochondrial alterations observed in LPS-activated macrophages are likely passive consequences of in vitro nitric oxide production and nitrite accumulation rather than requisite signals that drive macrophage function.

Lastly, it is important to explicitly state that our results do not discount a role for other aspects of mitochondrial function discrete from oxidative phosphorylation in regulating pro-inflammatory macrophage effector function. For example, the increase in membrane potential observed with 24-h Pam3 treatment aligns with previous studies demonstrating that increasing mitochondrial membrane potential can enhance IL-1β production via RET-driven superoxide production (Mills et al, 2016). Furthermore, even after only 4-h exposure to Pam3 or Poly I:C, macrophages exhibited a longer NADH fluorescence lifetime. This suggests redox changes could occur early in the pro-inflammatory response independently from alterations in oxidation phosphorylation or the mitochondrial membrane potential. As such, our findings suggest that the generation of relevant mitochondrial signals—such as TCA cycle-linked metabolites and redox triggers—is compatible with healthy oxidative phosphorylation and physiologically relevant bioenergetic parameters (Robb et al, 2018). Overall, the study supports a wide range of plasticity in the metabolic programs that can support pro-inflammatory macrophage activation rather than a uniform metabolic phenotype.

## Methods

### Reagents and tools table

| Reagent/resource | Reference or source | Identifier or catalog number |
|---|---|---|
| **Experimental models** | | |
| C57BL/6J (*M.musculus*) | The Jackson Laboratory | Strain# 000664 |
| B6.129P2(SJL)-Myd88<tm1.1Defr >/J (*M.musculus*) | The Jackson Laboratory | Strain# 009088 |
| C57BL/6J-Ticam1 < LPS2 >/J (*M.musculus*) | The Jackson Laboratory | Strain# 005037 |
| C57BL/6NJ-Acod1/J (*M.musculus*) | The Jackson Laboratory | Strain# 029340 |
| B6.129P2-Nos2<tm1Lau >/J (*M.musculus*) | The Jackson Laboratory | Strain# 002609 |
| *Ifnar*$^{-/-}$ (*M.musculus*) | Dr. Ting-Ting Wu, UCLA | N/A |
| Human PBMCs | UCLA CFAR Virology Core | N/A |
| J2-transformed immortalized BMDMs | Feng et al, 2024 | N/A |
| **Oligonucleotides and other sequence-based reagents** | | |
| *Il1b* primer: Fwd: GCCCATCCTCTGTGACTCAT Rev: AGGCCACAGGTATTTTGTCG | This paper | N/A |

| Reagent/resource | Reference or source | Identifier or catalog number |
|---|---|---|
| *Il6* primer: Fwd: AGTTGCCTTCTTGGGACTGA Rev: TCCACGATTTCCCAGAGAAC | This paper | N/A |
| *Tnfa* primer: Fwd: TGCCTATGTCTCAGCCTCTTC Rev: GAGGCCATTTGGGAACTTCT | This paper | N/A |
| *Nos2* primer: Fwd: CACCTTGGAGTTCACCCAGT Rev: ACCACTCGTACTTGGGATGC | This paper | N/A |
| *Il12b* primer: Fwd: ATCGTTTTGCTGGTGTCTCC Rev: GGAGTCCAGTCCACCTCTACA | This paper | N/A |
| *Irg1* primer: Fwd: GCAACATGATGCTCAAGTCTG Rev: TGCTCCTCCGAATGATACCA | This paper | N/A |
| *Mx1* primer: Fwd: GACCATAGGGGTCTTGACCAA Rev: AGACTTGCTCTTTCTGAAAAGCC | This paper | N/A |
| *Mx2* primer: Fwd: GAGGCTCTTCAGAATGAGCAAA Rev: CTCTGCGGTCAGTCTCTCT | This paper | N/A |
| *Ifnb1* primer: Fwd: CAGCTCCAAGAAAGGACGAAC Rev: GGCAGTGTAACTCTTCTGCAT | This paper | N/A |
| *Isg20* primer: Fwd: AACATCCAGAACAACTGGCGG Rev: GTCTGACGTCCCAGGGCA | This paper | N/A |
| *Irf1* primer: Fwd: GGCCGATACAAAGCAGGAGAA Rev: GGAGTTCATGGCACAACGGA | This paper | N/A |
| *Irf7* primer: Fwd: TCCAGTTGATCCGCATAAGGT Rev: CTTCCCTATTTTCCGTGGCTG | This paper | N/A |
| *Ifi204* primer: Fwd: CAGGGAAAATGGAAGTGGTG Rev: CAGAGAGGTTCTCCCGACTG | This paper | N/A |
| *Ndufs4* primer: Fwd: GTCTGTAGAGTTCCATCCAG Rev: GAGCAGGAACAAAGATTCTG | This paper | N/A |
| *36b4* primer: Fwd: CTGTGCCAGCTCAGAACACTG Rev: TGATCAGCCCGAAGGAGAAG | This paper | N/A |
| *Rosa26* sgRNA sequence: ACTCCAGTCTTTCTAGAAGA | This paper | N/A |
| *Ndufs4* sgRNA sequence #1: CUGGGCCCCGGAGGCGGCCU | This paper | N/A |
| *Ndufs4* sgRNA sequence #2: UGACAGUUAUGUUACCUGGA | This paper | N/A |
| *Ndufs4* sgRNA sequence #3: GAUGUUGGGGCGAAGGGCAA | This paper | N/A |

| Reagent/resource | Reference or source | Identifier or catalog number |
|---|---|---|
| **Chemicals, enzymes, and other reagents** | | |
| Oligomycin | Sigma-Aldrich | Cat# 75351 |
| Rotenone | Sigma-Aldrich | Cat# R8875 |
| Antimycin A | Sigma-Aldrich | Cat# A8674 |
| FCCP | Sigma-Aldrich | Cat# C2920 |
| Glucose | Sigma-Aldrich | Cat# G8769 |
| Sodium pyruvate | Sigma-Aldrich | Cat# P5280 |
| L-Glutamine | Sigma-Aldrich | Cat# G3126 |
| HEPES buffer | Sigma-Aldrich | Cat# 83264 |
| DMEM powder | Sigma-Aldrich | Cat# 5030 |
| Sodium chloride | Sigma-Aldrich | Cat# S7653 |
| Phenol Red | Sigma-Aldrich | Cat# P0290 |
| HCl | Sigma-Aldrich | Cat# 2104 |
| $H_2SO_4$ | Sigma-Aldrich | Cat# 1.60315 |
| Water suitable for cell culture | Sigma-Aldrich | Cat# W3500 |
| RBC lysis buffer | Sigma-Aldrich | Cat# R7757 |
| DMEM high-glucose | Gibco | Cat# 11965 |
| HyClone Characterized FBS | GE | Cat# SH30071.03 |
| Penicillin-streptomycin | Fisher Scientific | Cat# 15-140-122 |
| GlutaMAX | Fisher Scientific | Cat# 35-050-061 |
| Sodium pyruvate (100 mM) | Fisher Scientific | Cat# 11-360-070 |
| M-CSF | Takeshita et al, 2000 | N/A |
| Recombinant human M-CSF | PeproTech | Cat# 300-25 |
| Ultrapure LPS-Salmonella Minnesota | InvivoGen | Cat# TLRL-SMLPS |
| Pam3CSK4 | InvivoGen | Cat# TLRL-PMS |
| Poly(I:C) HMW | InvivoGen | Cat# TLRL-PIC |
| Imiquimod | InvivoGen | Cat# TLRL-IMQ |
| Recombinant murine IFN gamma | PeproTech | Cat# 315-05 |
| Recombinant mouse IFN beta1 | BioLegend | Cat# 581302 |
| Heat-killed staphylococcus aureus | InvivoGen | Cat# TLRL-HKSA |
| CL307 | InvivoGen | Cat# TLRL-C307 |
| ODN1668 | InvivoGen | Cat# TLRL-1668 |
| Piericidin A | Cayman Chemical | Cat# 15379 |
| Bam15 | Sigma-Aldrich | Cat# SML1760 |
| Carboxyatractolyside | Sigma-Aldrich | Cat# TA9H93ED6E11 |
| Cell-TAK | Fisher Scientific | Cat# CB-40240 |
| Seahorse XF Plasma Membrane Permeabilizer | Agilent | Cat# 102504-100 |
| Adenosine 5'-diphosphate (ADP) | Sigma-Aldrich | Cat# A5285 |
| Pyruvic acid | Sigma-Aldrich | Cat# 107360 |
| Malic Acid | Sigma-Aldrich | Cat# 240176 |

| Reagent/resource | Reference or source | Identifier or catalog number |
|---|---|---|
| Glutamic acid | Sigma-Aldrich | Cat# G1251 |
| Succinic acid | Sigma-Aldrich | Cat# S3674 |
| Citric acid | Sigma-Aldrich | Cat# 251275 |
| Mannitol | Sigma-Aldrich | Cat# M9546 |
| Sucrose | Sigma-Aldrich | Cat# 84097 |
| Egtazic acid (EGTA) | Sigma-Aldrich | Cat# E3889 |
| Bovine serum albumin (BSA) | Sigma-Aldrich | Cat# 2905 |
| Potassium hydroxide | Sigma-Aldrich | Cat# 221473 |
| Potassium phosphate monobasic | Sigma-Aldrich | Cat# P5655 |
| Magnesium chloride solution (1 M) | Sigma-Aldrich | Cat# 63069 |
| Dulbecco's phosphate-buffered saline (DPBS) | Fisher Scientific | Cat# 14-190-250 |
| 16% Paraformaldehyde | Fisher Scientific | Cat# 50-980-487 |
| Hoescht 33342 | Fisher Scientific | Cat# H3570 |
| Alamethicin from *Trichoderma viride* | Sigma-Aldrich | Cat# A4665 |
| Cytochrome c from bovine heart | Sigma-Aldrich | Cat# C2037 |
| β-Nicotinamide adenine dinucleotide, reduced dipotassium salt (NADH) | Sigma-Aldrich | Cat# N4505 |
| TMRE | Invitrogen | Cat# T669 |
| MitoTracker Green | Invitrogen | Cat# M7514 |
| Methanol | Sigma-Aldrich | Cat# 34860-4L-R |
| Chloroform | Sigma-Aldrich | Cat# 366927 |
| Norvaline | Sigma-Aldrich | Cat# N7502 |
| Methoxyamine hydrochloride | Sigma-Aldrich | Cat# 226904 |
| Pyridine | Sigma-Aldrich | Cat# 270407 |
| *N*-ter-Butyldimethylsilyl-*N*-methyl-trifluoroacetamide w/ 1% tert-butyldimethylchlorosilane | Sigma-Aldrich | Cat# 00942 |
| $^{13}C_6$-glucose | Cambridge Isotope Laboratories | Cat# CLM-1396-5 |
| $^{13}C_6$-glutamine | Cambridge Isotope Laboratories | Cat# CLM-1822-H-0.25 |
| Griess Reagent | Sigma-Aldrich | Cat# G4410 |
| Synthetic Guide RNAs | SYNTHEGO | N/A |
| Alt-R® Cas9 electroporation enhancer, 10 nmol | IDT | Cat# 1075916 |
| SpCas9 | SYNTHEGO | N/A |
| **Software** | | |
| Agilent Seahorse XF Wave | Agilent | https://www.agilent.com/en/product/cell-analysis/real-time-cell-metabolic-analysis/xf-software/seahorse-wave-pro-software-2007523 |

| Reagent/resource | Reference or source | Identifier or catalog number |
|---|---|---|
| Agilent GCMS MassHunter | Agilent | N/A |
| FluxFix | http://fluxfix.science | N/A |
| MetaXpress | Molecular Devices | N/A |
| FIJI | ImageJ, NIH | N/A |
| Attune NxT Software v3.1.2 | Thermo Fisher | N/A |
| GraphPad Prism | GraphPad | https://www.graphpad.com/ |
| Excel | Microsoft | https://www.microsoft.com/en-us/microsoft-365/excel |
| **Other** | | |
| Seahorse XFe96 Analyzer | Agilent | https://www.agilent.com/en/product/cell-analysis/real-time-cell-metabolic-analysis/xf-analyzers/seahorse-xfe96-analyzer-740879 |
| Seahorse XFe96 FluxPak | Agilent | 102416-100 |
| Operetta CLS High-Content Analysis System | PerkinElmer | N/A |
| Multimode plate reader | Tecan | N/A |
| ImageXpress | Molecular Devices | N/A |
| LSM880 | Zeiss | N/A |
| CentriVap vacuum concentrator | LabConco | N/A |
| Attune NxT Flow Cytometer | Thermo Fisher | N/A |
| RNeasy Mini Kit | Qiagen | Cat# 74106 |
| High-capacity cDNA reverse transcription kit | Applied Biosystems | Cat# 4368814 |
| PowerUP SYBR Green qPCR Master Mix | Applied Biosystems | Cat# A25743 |
| QuantStudio 5 | Applied Biosystems | N/A |
| ELISA MAX Deluxe Set Mouse IL-6 | BioLegend | Cat# 431304 |
| ELISA MAX Deluxe Set Mouse IL-12/IL-23 (p40) | BioLegend | Cat# 431604 |
| Luminex xMAP® Immunoassay | Thermo Fisher | N/A |
| LEGENDplex Mouse M1 Macrophage Panel (8-plex) with V-bottom plate | BioLegend | Cat# 740848 |
| Phagocytosis Assay Kit (Green E.coli) | Abcam | Cat# AB235900 |
| Neon Transfection system | Thermo Fisher | Cat# MPK5000 |
| Gene knockout sgRNAt kit targeting the *rosa26* locus | SYNTHEGO | N/A |

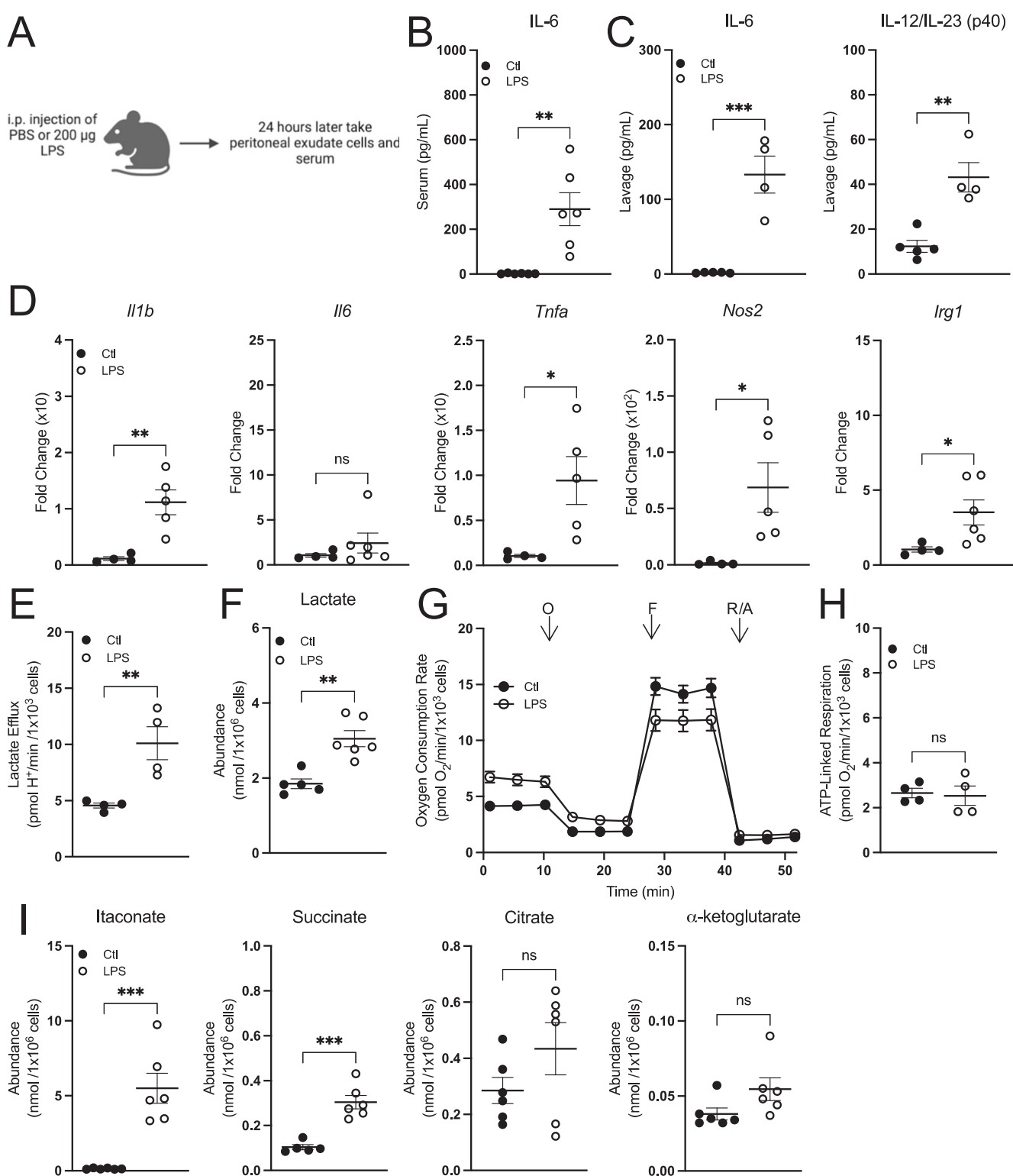

Animals

## Animals

Animal housing and experimental procedures were authorized by the UCLA Animal Research Committee under protocol ARC 2020-027. Mice were housed four per cage in a temperature—(22–24 °C)

and humidity-controlled colony room maintained on a 12 h light/dark cycle (07:00 to 19:00 light on). They were fed a standard chow diet (LabDiet 5053), and water was provided ad libitum with environmental enrichments. The general health of the animal was assessed weekly by UCLA veterinarians.

**Figure 7. Peritoneal macrophages activated in vivo preserve oxidative phosphorylation and accumulate pro-inflammatory metabolites.**

(A) Schematic depicting the experimental design for peritoneal macrophages. i.p. intraperitoneal. (B, C) Cytokine levels from the harvested serum (B) ($n = 5$–6) and lavage fluid (C) ($n = 4$–6) after 24 h from mice intraperitoneally injected with 200 µg LPS or PBS as a control (Ctl) (B: **$P = 0.003$; C: ***$P < 0.001$, **$P = 0.002$). (D) Pro-inflammatory gene expression from peritoneal macrophages isolated from mice treated as in (A–C) ($n = 4$–6) (*Il1b*: **$P = 0.005$; *Tnfa*: *$P = 0.03$; *Nos2*: *$P = 0.03$; *Irg1*: *$P = 0.05$). (E) Lactate efflux rates for peritoneal macrophages isolated from mice treated as in (A–C) ($n = 4$) (**$P = 0.01$). (F) Intracellular lactate levels from peritoneal macrophages isolated from mice treated as in (A–C) ($n = 5$–6) (**$P = 0.001$). (G) Representative oxygen consumption traces with peritoneal macrophages isolated from mice treated as in (A–C) ($n = 1$ biological replicate with 5 technical replicates). (H) Aggregate ATP-linked respiration rates for peritoneal macrophages isolated from mice treated as in (A–C) ($n = 4$). (I) Intracellular itaconate, succinate, citrate, and α-ketoglutarate levels from peritoneal macrophages isolated from mice treated as in (A–C) ($n = 6$) (***$P < 0.001$). All data presented in Fig. 7 are mean ± standard error of the mean (SEM) with statistical analysis conducted on data from biological replicates, each of which included multiple technical replicates, unless otherwise indicated. When not visible, error bars are obscured by the symbol for all oxygen consumption traces. O oligomycin, F FCCP, R/A rotenone/antimycin A. Statistical analysis for (B–F) and (H, I) was performed as an unpaired, two-tailed *t*-test. Source data are available online for this figure.

The following strains were purchased from The Jackson Laboratory: C57BL/6J (strain #000664); B6.129P2(SJL)-Myd88<tm1.1Defr > /J (strain #009088); C57BL/6J-Ticam1 < LPS2 > /J (strain #005037); C57BL/6NJ-Acod1/J (strain #029340); B6.129P2-Nos2<tm1Lau > /J (strain #002609). Femurs and tibias from *Ifnar$^{-/-}$* mice were generously provided by Dr. Ting-Ting Wu.

## Isolation of mouse bone marrow-derived macrophages (BMDMs)

Bone marrow cells were isolated from femurs of male mice between the age of 8–12 weeks, as previously described (Hsieh et al, 2021). Briefly, cells were treated with 3 mL RBC lysis buffer (Sigma-Aldrich) for 4 min to remove red blood cells, centrifuged at $400 \times g$ for 5 min, and resuspended in the cell culture medium described below. Cells were maintained at 37 °C in a humidified 5% CO$_2$ incubator. BMDMs were differentiated for 6 days prior to experimental treatments, and the medium was changed on day 4 of differentiation.

For all experiments involving BMDMs, cells were cultured in high-glucose DMEM (Gibco 11965) supplemented with 10% (v/v) heat-inactivated fetal bovine serum (FBS) unless otherwise indicated, 2 mM L-glutamine, 100 units/mL penicillin, 100 µg/mL streptomycin, 500 µM sodium pyruvate, and 5 or 10% (v/v) conditioned media containing macrophage colony-stimulating factor (M-CSF) produced by CMG cells to induce differentiation to BMDMs (Takeshita et al, 2000).

## Isolation of human PBMC-derived macrophages

Single donor human peripheral blood mononuclear cells (PBMCs) were obtained from the UCLA CFAR Virology Core lab for monocyte isolation. Monocytes were subsequently prepared by standard Ficoll isolation procedures and plastic adherence (Zhou et al, 2020). For macrophage differentiation, $1 \times 10^7$ monocytes were suspended in DMEM supplemented with 10% (v/v) FBS with 2 mM L-glutamine, 100 units/mL penicillin, 100 µg/mL streptomycin, 500 µM sodium pyruvate, and 50 ng/mL of recombinant human M-CSF (Peprotech, 300-25) and cultured on non-tissue cultured treated Petri Dishes (Fisher #FB0875712) for 6 days prior to assays. The medium was changed on day 4 of differentiation.

## Isolation of mouse peritoneal macrophages

Mice were intraperitoneally injected with PBS, 200 µg LPS, or 200 µg Pam3 24 h prior to isolation of peritoneal macrophages. Researchers

were not kept blind to the treatment groups. Mice were euthanized with isoflurane, followed by cervical dislocation, and the abdominal skin was retracted to expose the intact peritoneal wall. 5 mL of ice-cold PBS with 2 mM EDTA and 2% (v/v) fetal calf serum (Biochrom) was injected into the peritoneal cavity using a syringe with a 20-G needle. Following gentle massages of the cavity, the fluid was then aspirated from the peritoneal cavity using the same syringe and collected in a 15 mL tube. The procedure was repeated twice to obtain a final volume of 10 mL. The cell suspension was centrifuged at $400 \times g$ for 5 min. Cell pellets were resuspended in BMDM culture medium and plated in 12-well plates. Cells were incubated for 3 h at 37 °C in a humidified 5% CO$_2$ incubator with peritoneal macrophages adhering to the plate and other cells remaining in the supernatant. For peritoneal macrophages isolated for in vitro studies, mice were intraperitoneally injected with 3 mL of sterile thioglycolate broth for 72 h prior to isolation.

## Stimulation of bone marrow-derived macrophages

On day 6 after harvest, BMDMs were plated at different densities per well for the respective assays (see below). On day 8, macrophages were treated with 50 ng/mL LPS (or 10 ng/mL as noted in the figure legend), 50 ng/mL Pam3CSK4, 1 µg/mL Poly I:C, 10 µM imiquimod, 20 ng/mL IFN-γ, 20 ng/mL IFN-β, $10^7$ cells/mL heat-killed staphylococcus A (HKSA), 100 nM CL307, 100 nM ODN1668 or simultaneously co-stimulated with a combination of the above for 24 h alongside matched vehicle controls. For the 4-h treatments, on day 8, the medium was changed, and on day 9, macrophages were stimulated as noted in the figure legends.

## Mitochondrial effector compound treatment of BMDMs

For experiments involving respiratory chain inhibitors, BMDMs were treated with vehicle controls or with 100 nM piericidin A, 30 nM antimycin A, 10 nM oligomycin, 10 nM oligomycin + 3 µM Bam15, or 30 µM carboxyatractyloside (CAT) for 24 h. All inhibitors were given as co-treatments simultaneously with 50 ng/mL Pam3CSK4, 1 µg/mL Poly I:C, or 10 ng/mL LPS.

## In vitro peritoneal macrophage stimulation

Freshly isolated, thioglycolate-elicited peritoneal macrophages were seeded in Seahorse XF96 wells at a seeding density of $5 \times 10^4$ cells/well in BMDM culture medium. The following day, cells were treated with either 50 ng/mL LPS or 50 ng/mL Pam3, and respiration was assessed after 24 h.

## Seahorse XF analysis

All respirometry was conducted in a Seahorse XF96 or XFe96 Analyzer (Agilent Technologies). All experiments were conducted at 37 °C and at pH 7.4 (intact cells) or 7.2 (permeabilized cells). The placement of treatment groups on the XF plate was randomized across biological replicates as best as possible to avoid biased results.

### Intact cells

Respiration was measured in a medium containing 8 mM glucose, 2 mM glutamine, 2 mM pyruvate, and 5 mM HEPES. BMDMs were plated at $5 \times 10^4$ cells/well on day 6 and assayed on day 9 after treatments as described in figure legends. iBMDMs were plated at $2.5 \times 10^4$ cells/well for 48 h prior to measurement, with pro-inflammatory stimuli added 24 h after plating. Respiration was measured in response to oligomycin (1 μM), carbonyl cyanide-$p$-trifluoromethoxyphenylhydrazone (FCCP) (0.75 nM or 1.5 μM), and rotenone (0.2 μM) with antimycin A (1 μM). Calculations of respiratory parameters were made according to standard protocols (Divakaruni et al, 2014; Divakaruni and Jastroch, 2022). Briefly, ATP-linked respiration was calculated by subtracting the oxygen consumption rate insensitive to oligomycin from the initial measurements. Maximal respiration was calculated by subtracting the oxygen consumption rate insensitive to rotenone and antimycin A from the maximum rate obtained after injection of FCCP. Lactate efflux rates were calculated as previously described (Desousa et al, 2023).

For ex vivo peritoneal macrophages, cells were plated at $2.5 \times 10^5$ cells/well in Cell-Tak-coated (Corning) 96-well Seahorse XFe96 plates. Plates were spun at $500 \times g$ for 4 min, and respiratory parameters were obtained as previously indicated.

### Permeabilized and "double-permeabilized" cells

Recombinant, mutant perfringolysin O (rPFO; commercially XF Plasma Membrane Permeabilizer [XF PMP, Agilent Technologies]) was used to selectively permeabilize the plasma membrane of BMDMs. Experiments were conducted as previously described (Yang et al, 2021; Divakaruni et al, 2018a). Immediately prior to the assay, cell medium was replaced with MAS buffer (70 mM sucrose, 220 mM mannitol, 10 mM $KH_2PO_4$, 5 mM $MgCl_2$, 2 mM HEPES, 0.2% (w/v) Fraction V BSA, and 1 mM EGTA; pH 7.2) containing 3 nM rPFO, respiratory substrates, and 4 mM ADP. The ADP-stimulated respiration rate (referred to as "State 3" respiration) was measured, and background signal was measured after treatment with 0.2 μM rotenone with 1 μM antimycin A. Substrate concentrations were as follows: glutamate/malate, 5 mM glutamate with 5 mM malate; pyruvate/malate, 5 mM pyruvate with 1 mM malate; succinate/rotenone, 5 mM succinate with 2 μM rotenone; citrate, 5 mM citrate.

To directly assess complex I- and complex II-mediated respiration, rPFO-permeabilized iBMDMs were treated with alamethicin (10 μg/mL) to form pores of 3–6 kDa in the mitochondrial inner membrane ("double-permeabilized" cells). Alamethicin was added 15 min prior to measurements at 37 °C (Divakaruni et al, 2018a). "Double-permeabilized cells" were offered 10 μM cytochrome c in the experimental medium and either 10 mM NADH (complex I substrate) or 10 mM succinate (complex II substrate) with 2 μM rotenone to drive respiration.

## Mitochondrial membrane potential

BMDMs were plated in the inner 60 wells of black-walled, 96-well plates at $3 \times 10^4$ cells/well. Prior to measurements of membrane potential, the medium was changed to DMEM lacking serum, sodium bicarbonate, and antibiotic but supplemented with 10 nM TMRE (Invitrogen #T669), 200 nM MitoTracker Green (MTG; Invitrogen #M7514), and 1 μg/mL Hoechst (Thermo Fisher #33342). Cells were incubated in this medium for 1 h at 37 °C. After incubation, cells were washed two times with a similar incubation medium but lacking phenol red. Images were acquired using the ImageXpress instrument (Molecular Devices) with 50 mm slit confocal mode and a 40x (1.2 NA) water lens in Z-stack mode of 0.5 mm slices with a total of six slices. Analysis was performed in the MetaXpress software keeping the same parameters for all the images acquired. Maximum Z-projections of MTG were used for morphologic analysis and the sum of Z-projections of TMRE was used for quantification of intensity. A TopHat filter was applied to the MTG images for better definition of structures and equalization of fluorescence. The images were thresholded and transformed into a binary segmentation. This segmented area was used to measure the average intensity of TMRE on pixels positive for MTG.

## Mitochondrial morphology measurements

Imaging of BMDMs for morphometric analysis was conducted on a Zeiss LSM880 using a 63x Plan-Apochromat oil-immersion lens and AiryScan super-resolution detector with a humidified 5% $CO_2$ chamber on a temperature-controlled stage at 37 °C. Cells were differentiated in glass-bottom confocal plates (Greiner Bio-One). BMDMs were incubated with 15 nM TMRE for 1 h in their regular culture medium. For this experiment, TMRE was only used as a marker for mitochondria, and intensity was not quantified but rather morphometric analysis was conducted for mitochondria positive for TMRE staining. Image Analysis was conducted using FIJI (ImageJ, NIH). Image contrast and brightness were not altered in any quantitative image analysis protocols. Brightness and contrast were equivalently modified across measurement groups to allow proper representative visualization of the effects revealed by unbiased quantitation.

## Fluorescence lifetime imaging microscopy (FLIM) of endogenous NADH

Image acquisition was performed using a Zeiss LSM880 Confocal system equipped with a Zeiss Plan-Apochromat 63x/1.4 N.A. oil-immersion objective. Endogenous NADH was exited at 820 nm with a pulsed two-photon laser, and its emission was recorded by time-correlated single-photocounting (TCSPC) for 80 s using a 420–480 nm Bandpass filter in a hybrid PMT system (HPM-100-40 Becker & Hickl, GmbH). Fluorescence decay curves were fitted by the SPCImage analysis software with a bi-exponential decay function, including the instrument response function (IRF). Lifetime images were analyzed in ImageJ using ROIs displaying a whole cell. $\tau_{cell}$ describes the average NADH lifetime of single cells.

## Metabolite quantification with GC/MS analysis

GC/MS analysis was performed as previously described (Cordes and Metallo, 2019). Briefly, BMDMs were plated at $1 \times 10^6$ cells/

well in six-well plates and treated with macrophage stimuli as described earlier in the methods. Peritoneal macrophages were extracted immediately following the 3-h incubation in 12-well plates. Metabolite extraction was conducted with a Folch-like protocol with a 5:2:5 ratio of methanol:water:chloroform. 6- or 12-well dishes were kept on ice and quickly washed with ice-cold 0.9% (w/v) NaCl. Cells were then scraped in ice-cold methanol and water containing 5 µg/mL norvaline (Sigma #N7502) as an internal standard. Chloroform was then added, and samples were vortexed for 1 min and centrifuged at 10,000×g for 6 min at 4 °C. The order in which samples were harvested were randomized across biological replicates as best as possible to avoid biased results.

The polar fractions (top layer) of the samples were dried overnight using a refrigerated CentriVap vacuum concentrator (LabConco). Metabolite standards (50 nmol to 23 pmol) were extracted alongside the cell samples to ensure the signal fell within the linear detection range of the instrument. The dried polar metabolites were reconstituted in 20 µL of 2% (w/v) methoxyamine in pyridine prior to a 45-min incubation at 37 °C. Subsequently, 20 µL of *N*-tertbutyldimethylsilyl-*N*-methyl-trifluoroacetamide with 1% tert-butyldimethylchlorosilane was added to each sample, followed by an additional 45-min incubation at 37 °C. Samples were analyzed using a DB-35 column (Agilent Technologies) and quantified using Agilent MassHunter software (version 6.02). Information regarding additional technical specifications is available elsewhere (Vacanti et al, 2014; Cordes and Metallo, 2019).

## Stable isotope tracing

On day 6 of the BMDM culture, cells were seeded $1 \times 10^6$ cells/well in six-well plates in culture medium. When assessing the effect of 24-h treatment with stimuli, cells were treated on day 8 with ligands as indicated in the figure legend. Eighteen hours later, on day 9, the medium was changed to culture medium with ligands and either 10 mM uniformly labeled $^{13}C_6$-glucose (Cambridge Isotope Laboratories), or 6 mM uniformly labeled $^{13}C_5$-glutamine (Cambridge Isotope Laboratories). A similar strategy was previously used successfully so that the stable isotope tracer could be added after any pro-inflammatory stimuli was able to have an effect (Palmieri et al, 2020a). For the medium containing a given labeled metabolite, all other metabolites were still present at the same concentration and unlabeled (Cordes and Metallo, 2019). After a 6-h incubation with isotope tracers, metabolites were extracted as described above. FluxFix software (http://fluxfix.science) was used to correct for the abundance of natural heavy isotopes against an in-house reference set of unlabeled metabolite standards (Trefely et al, 2016). When assessing the effect of 4-h treatment with stimuli, the medium was changed on day 8 with a regular culture medium, and cells were treated for 4 h on day 9 with stable isotope tracers and pro-inflammatory stimuli as before prior to extraction.

## Quantitative real-time RT-PCR (qPCR)

For gene expression analysis, day 6 BMDMs were seeded at $3 \times 10^5$ cells/well in 12-well plates in BMDM culture medium. For 24-h treatments using BMDMs, cells were treated on day 8 with ligands as indicated in the figure legend in culture medium supplemented with 5% (v/v) FBS. For 4-h treatments using BMDMs, the medium was changed on day 8 with regular culture medium, and on day 9, cells

were treated with compounds as indicated in the figure legend in culture medium supplemented with 5% (v/v) FBS. iBMDMs were seeded at $2.5 \times 10^4$ cells/well for 24 h and treated with pro-inflammatory stimuli 24 h prior to lysis. Peritoneal macrophages were lysed immediately following the 3-h incubation in 12-well plates. All cell types were collected in Qiagen RNeasy Cell Lysis Buffer and RNA was extracted according to the manufacturer's protocol (Qiagen). cDNA was synthesized using 1000 ng RNA per reaction with a high-capacity cDNA reverse transcription kit (Applied Biosystems). KAPA SYBR Green qPCR Master Mix (2X) Kit (Applied Biosystems) and an Applied Biosystems QuantStudio 5 were used for quantitative RT-PCR using 0.5 µmol/L primers. Fold change relative to internal controls was calculated using the $2^{\Delta\Delta CT}$ method with *36b4* as the reference gene.

## Griess assay

Nitric oxide was measured from the BMDM culture medium 24 h after effector treatments. Cell-culture supernatants were centrifuged at 500×g for 5 min. to remove particulates. Nitrite, a stable product of nitric oxide degradation, was measured by mixing 50 µL of culture supernatants with 50 µL Griess reagent (Sigma #G4410), incubating in the dark for 15 min at room temperature, and measuring absorbance at 540 nm. A standard curve was constructed with sodium nitrite standards for absolute quantification of nitrite.

## Cytokine measurements

Enzyme-linked immunosorbent assays (ELISAs) were used to measure mouse IL-6 and IL-12b/IL-23 (p40) in BMDM culture medium (supernatant collected after centrifugation at 500×g for 5 min.), mouse serum, or mouse lavage fluid according to manufacturer's instructions (BioLegend). IL-1β levels in BMDM cell-culture supernatant was measured by the xMAP® Immunoassay (Luminex) with the UCLA Immune Assessment Core Facility. CXCL1 (KC), TNF-α, and IL-12p40 levels in BMDM cell-culture supernatant was measured using the LEGENDplex MU M1 Macrophage Panel (8-plex; BioLegend) with a V-bottom plate according to the manufacturer's instructions.

## Phagocytosis

In vitro phagocytosis of heat-killed, fluorescently pre-labeled *E. coli* particles in BMDMs was measured via flow cytometry according to manufacturer's instructions (Abcam #AB235900).

## CRISPR-mediated deletion of *Ndufs4* in immortalized macrophages

J2-transformed immortalized BMDMs were generated as previously described (Feng et al, 2024). A CRISPR multi-guide gene knockout kit was used to target the *Ndufs4* locus as well as a negative control guide targeting *Rosa26* (Synthego). The multi-guide sgRNA sequences targeting *Ndufs4* were 5′-CUGGGCCCCGGAGGC GGCCU-3′, 5′-UGACAGUUAUGUUACCUGGA-3′, and 5′-GAU-GUUGGGGGCGAAGGGCAA-3′. cRNP-mediated editing of immortalized BMDMs (iBMDMs) was performed as previously described (Hildreth et al, 2020). Briefly, 120 pmol of Synthetic sgRNA (Synthego) and 9 pmol of Alt-R electroporation enhancer

(Integrated DNA Technologies) were mixed with 20 pmol of SpCas9 (Synthego). This mixture was incubated for 25 min at room temperature to form a cRNP complex. iBMDMs were then electroporated with cRNP complexes using a Neon Transfection system (Thermo Fisher) with 1900 V and a pulse width of 20 ms. Electroporated cells were allowed to recover for 90 min and placed in BMDM culture medium. After 3 days, the CRISPR-edited cells were used for downstream assays.

## Cell counts and normalization

When normalizing respirometry experiments and metabolite quantification to cell number, BMDMs were fixed immediately upon completion of the assay with 2% (w/v) formaldehyde for 20 min at room temperature and kept refrigerated between 1 and 14 days until assessment. On the day prior to cell counting, cells were stained with Hoechst (Thermo Fisher #33342) at 10 ng/mL overnight at room temperature. Cell counts were obtained using the Operetta High-Content Imaging System (PerkinElmer).

## Statistical analysis

All statistical parameters, including the number of biological replicates ($n$), can be found in the figure legends. Each data point represents a biological replicate obtained from an individual mouse/human sample and is comprised of the average of multiple technical replicates unless explicitly stated otherwise. Statistical analyses were performed using GraphPad Prism 10 software. Data were presented as the mean ± standard error of the mean (SEM). Individual pairwise comparisons were performed using a two-tailed Student's $t$-test. For experiments involving two or more factors, data were analyzed by one-way, repeated measures ANOVA followed by Tukey's post hoc multiple comparisons tests. For other multiple values comparisons, data were analyzed by ordinary two-way, ANOVA followed by Tukey's or Sídák's post hoc multiple comparisons test when required. Data were assumed to follow a normal distribution (no tests were performed). Values denoted as follows were considered significant: *$p < 0.05$; **$p < 0.002$; ***$p < 0.001$.

## Data availability

This study has neither produced nor deposited novel datasets.

The source data of this paper are collected in the following database record: biostudies:S-SCDT-10_1038-S44319-024-00351-y.

## Peer review information

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

## Acknowledgements

ASD is supported by the National Institutes of Health (NIH) Grant R35GM138003, the W.M. Keck Foundation (995337), and the Agilent Early Career Professor Award (4814). ABB is supported by the UCLA Chemistry-Biology Interface Training Program (T32 GM136614). AEJ was supported by the UCLA Tumor Cell Biology Training Program (T32CA009056). SJB is supported by the NIH Grants P01HL146358 and R01HL157710. We acknowledge the UCLA/CFAR Virology Core Lab (5P30 AI028697), the UCLA Immune Assessment Core Facility, and the Metabolism and Mitochondria Bioenergetics and Imaging Cores at UCLA for their assistance in sample and data collection. Graphics in Figs. 1A, D, 3A, 7A; Appendix Fig. S5A were created with BioRender.com.

## Author contributions

**Andréa B Ball**: Data curation; Formal analysis; Investigation; Writing—original draft; Writing—review and editing. **Anthony E Jones**: Data curation; Formal analysis; Investigation; Methodology. **Kaitlyn B Nguyễn**: Data curation; Formal analysis; Writing—review and editing. **Amy Rios**: Data curation. **Nico Marx**: Data curation; Formal analysis. **Wei Yuan Hsieh**: Data curation; Formal analysis; Investigation; Methodology. **Krista Yang**: Data curation; Formal analysis. **Brandon R Desousa**: Data curation; Formal analysis. **Kristen KO Kim**: Data curation; Formal analysis. **Michaela Veliova**: Data curation; Formal analysis. **Zena Marie del Mundo**: Data curation. **Orian S Shirihai**: Conceptualization; Resources. **Cristiane Benincá**: Data curation; Formal analysis; Methodology. **Linsey Stiles**: Resources; Data curation; Formal analysis; Methodology. **Steven J Bensinger**: Conceptualization; Resources; Funding acquisition; Methodology. **Ajit S Divakaruni**: Conceptualization; Resources; Data curation; Formal analysis; Supervision; Funding acquisition; Investigation; Methodology; Writing—original draft; Project administration; Writing—review and editing.

Source data underlying figure panels in this paper may have individual authorship assigned. Where available, figure panel/source data authorship is listed in the following database record: biostudies:S-SCDT-10_1038-S44319-024-00351-y.

## Disclosure and competing interests statement

The authors declare no competing interests.

