## [Peer Review File · EMBO Reports]

Pro-inflammatory macrophage activation does not require inhibition of oxidative phosphorylation

Andréa Ball, Anthony Jones, Kaitlyn Nguyễn, Amy Rios, Nico Marx, Wei Hsieh, Krista Yang, Brandon Desousa, Kristen Kim, Michaela Veliova, Zena Marie del Mundo, Orian Shirihai, Cristiane Benincá, Linsey Stiles, Steven J. Bensinger, and Ajit Divakaruni

Corresponding author(s): Ajit Divakaruni (ADivakaruni@mednet.ucla.edu)

Review Timeline:	Transfer Date:	18th Sep 24
	Editorial Decision:	21st Oct 24
	Revision Received:	4th Dec 24
	Accepted:	5th Dec 24

Editor: Esther Schnapp

Transaction Report: Please note that the manuscript was transferred from another journal where it was originally reviewed. Since the original reviews are not subject to EMBO's transparent review process policy, they cannot be published.

RESPONSE TO REVIEWERS

Our remarks in **black**; reviewer comments in *red italics*

We would like to genuinely thank the reviewers for their thorough and thoughtful comments. The suggestions have made our manuscript **much** better, and made it clear we erred in both the framing and depth with which we addressed the question. Prior to the point-by-point response, we ask the reviewers please note the large-scale changes made to the manuscript as a result of the peer review:

- (1) Our revised manuscript is now a full-length article with 7 main figures which we now submit to *EMBO Reports* rather than a brief, 4-figure letter to *Nature Metabolism*.
- (2) In addition to the expanded format, we have also completely changed the sequence of how our results are presented. Rather than begin with a comparison of the pro-inflammatory response at 4 vs. 24 hours, we now begin with the demonstration that not all pro-inflammatory stimuli inhibit oxidative phosphorylation, and reserve a more in-depth kinetic analysis comparing gene expression with respiration until Figure 6.
- (3) Our manuscript seeks to rigorously test the conventional model that reductions in oxidative phosphorylation (not necessarily all mitochondrial function as a whole) amplify pro-inflammatory macrophage activation. As such, we focused most of our initial attention on respiration, specifically oxygen consumption coupled to ATP synthesis. We are grateful that the reviewers suggested we use orthogonal measurements of mitochondrial function, as they have strengthened our conclusions. We now include measurements of (i) protein-bound NADH with fluorescence lifetime imaging (FLIM), (ii) stable isotope tracing with $^{13}\text{C}_6$ -glucose and $^{13}\text{C}_5$ -glutamine, (iii) permeabilized cell respirometry, and (iv) higher-resolution measurements of the mitochondrial membrane potential to add to our existing analysis of respiration, metabolite abundance, and mitochondrial morphology.

Reviewer #1 (Remarks to the Author):

Major points related to Figure 1

1, Line 315-316, the title of Figure 1: “Mitochondrial respiration is unchanged when pro-inflammatory gene expression nears its peak”; would be better if it’s been rephrased? The claim is not entirely supported by the evidence provided, for several reasons:

This is a very helpful point. We have stripped such inexact language from the manuscript as a whole, restructured the manuscript to put a time course later in the data presentation, and experimentally addressed the suggestions as detailed below:

A) The Seahorse Cell Mito Stress Test is a common method to measure the key parameters of mitochondrial respiration (yet not sensitive enough to detect delicate changes in the current scenario, let alone being “conclusive” enough in general). Other key parameters, such as mitochondrial membrane potential and mitochondrial NAD(P)H should be considered as well. It would be more convincing to have data on $\Delta\psi_m$ and NAD(P)H ratios (though they might not be influenced entirely by mitochondrial respiration but a check of them and further discussion is necessary). More conclusive would be ^{13}C -glucose tracing experiments (more proper to have this set data in Figure 3, details in point #6) and those addressing the two proposed “break points”, mitochondrial aconitase (ACO2)/Isocitrate dehydrogenase (IDH) and succinate dehydrogenase (SDH) activity in vitro. Please either provide further evidence mentioned above or a discussion if the data don’t support the claim.

This suggestion in particular has greatly improved our piece and strengthened our findings. We have conducted the following experiments, all of which reinforce our central finding that reductions in oxidative phosphorylation are not obligatory for macrophage pro-inflammatory activation.

- **Stable isotope tracing:** We have conducted stable isotope tracing with uniformly labeled $^{13}\text{C}_6$ -glucose and $^{13}\text{C}_5$ -glutamine after both 4 hr. and 24 hr. treatment of BMDMs with various pro-inflammatory stimuli. **(Figs. 4 & 6)**

- Mitochondrial membrane potential: We have also conducted high-resolution measurements of the mitochondrial membrane potential [TMRE (10 nM)] normalized to mitochondrial content [MitoTracker Green (200 nM)] after both 4 hr. and 24 hr. treatment with pro-inflammatory stimuli. (**Figs. 1 & 6**)
- NADH fluorescence lifetime imaging measurement (FLIM): We have also conducted FLIM to assess protein-bound NADH after both 4 hr. and 24 hr. treatment with pro-inflammatory stimuli. (**Figs. 1 & 6**)
- Permeabilized cell respirometry: Additionally, we have conducted permeabilized cell respirometry to provide more in-depth analysis about specific metabolic pathways after both 4 hr. and 24 hr. treatment with pro-inflammatory stimuli. (**Figs. 4 & 6**)

B) The change of mitochondrial respiration after LPS stimulation has been shown to be continuous and gradual at the early phase, so it seems to me improper to claim it's totally "unchanged".

This is a good point and we have tried to treat our data with appropriate nuance in the revision. In addition to being more precise with our wording throughout the piece, we have included a more thorough time course measuring pro-inflammatory gene expression alongside respiratory inhibition as detailed below in the response to 1C.

C) Pro-inflammatory genes expression does not seem to be simultaneous or synchronized; and the kinetic data of these pro-inflammatory genes aren't quite adequate to claim 4h is "near its peak". The current data set indicates quite a few peaks at around 8h, by which time I believe the mitochondrial respiration has been changed.

We agree it was inappropriate to make claims about kinetic effects without a proper time course. As such, **Figs. 6 and S9** now show a time course matching gene expression of several genes with maximal respiration rates at time points of 1, 2, 3, 4, 5, 6, 8, and 12hr. Expression of many key pro-inflammatory genes peaks between 3-5 hours. At these times, we do not observe a difference in maximal oxygen consumption relative to vehicle-treated cells, and respiratory capacity actually increases early after stimulation with LPS. Coupled with our data at the 4 hr. timepoint showing there are no substantive alterations in mitochondrial membrane potential, fluorescence lifetime imaging on NADH, permeabilized cell respirometry, and ¹³C stable isotope tracing, we provide several lines of evidence that mitochondrial bioenergetics are not compromised during the early induction of the pro-inflammatory response.

2, Supplementary figure 1A, does "Pam3" in the graph mean anything?

We apologize for the oversight in labeling. 'Pam3' refers to Pam3-CSK4, a tri-acylated lipopeptide that is used as a specific agonist for TLR2.

3, Line 129, a brief explanation or discussion of IL1 β and TNF α release (Western Blot of the cell lysis if it's undetectable in medium) would possibly be appreciated since most previous, influential studies have focused on them, particularly IL1b.

We now do a better job of reinforcing our measurements of gene expression with cytokine secretion. We have restructured the focus of our manuscript away from the 4 hr. timepoint as a central finding, so addressing the comment relative to what was formerly Line 129 is no longer applicable. However, we have added additional measurements of IL-1 β and TNF- α secretion to **Fig. 5**, demonstrating that inhibition of the respiratory chain does not augment TLR2/MyD88-driven secretion of these cytokines.

Major points related to Figure 2

4, Figure 2 ABCDEHI, what was the time of LPS or Pam3 (+-PIC) stimulation (presumably 24h for all)? Particularly H&I, please specify the IFN- γ dose (and later IFN β).

We thank the reviewer for pointing out the confusion in switching between time points so abruptly between Figures 1 & 2 of the previous manuscript iteration. We have now restructured the piece so Figures 1-5 are all 24 hr. treatments followed by a more thorough time course analysis presented after in Fig. 6. Additionally, we have now made clear in the methods and figure legends that all interferon treatments (IFN- β or IFN- γ) were used at 20 ng/mL.

Major points related to Figure 3

5, Line 195, “accumulation of ... itaconate and succinate occurs independently from the inhibition of oxidative phosphorylation”; this is quite a surprise. It was logical to assume that the accumulation of itaconate and succinate would be the consequence of TCA cycle “breaks”. A comment, explanation discussing the possible mechanisms of these accumulations would be good? Is it possible that these breaks happened, but oxygen consumption wasn’t affected yet? Are the major enzymes still functioning normally at this point? What are the sources of the increased levels of itaconate and succinate, glucose, glutamine, or lipid?

We were constrained by the previous “Letter” format from discussing this finding in-depth, and are now able to do so in a full-length manuscript. We have expanded our results showing that accumulation of succinate and itaconate is discrete from respiratory inhibition over two figures in **Figures 3 & 4**. Each of the discrete points from the comment are discussed here:

- Why do itaconate and succinate not accumulate as a consequence of ‘breaks’ in the TCA cycle?
 - One of the more surprising findings of our study is that respiratory inhibition is discrete from accumulation of itaconate and succinate. We show this both with (i) itaconate accumulation in treatments that do not inhibit respiration, and (ii) profound respiratory inhibition even in the absence of itaconate accumulation in BMDMs from *Irg1*^{-/-} mice.

The steady-state concentration of any metabolite is a representation of both its generation (‘flux in’) and its consumption (‘flux out’). As such, it is possible to increase steady-state levels of itaconate by simply readjusting the balance of production and consumption.

There are several instances in both physiology and disease in which metabolite generation does not occur hand-in-hand with respiratory inhibition. For example, lipogenesis requires an increase in citrate siphoned out of the TCA cycle to fuel ATP citrate lyase. Additionally, the generation of millimolar quantities of 2-hydroxyglutarate from gain-of-function IDH1/2 mutations does not alter oxidative phosphorylation in either normoxia or hypoxia [Grassian et al (2014) *Cancer Research*].

Our data suggest that induction of *Irg1* itself – independent of inhibition of IDH or other mitochondrial dehydrogenases – is sufficient to generate itaconate. Treatments that do not collapse oxidative phosphorylation, such as Pam3 (TLR2 agonist) or Poly I:C (TLR3), simply redirect carbon towards itaconate accumulation without changing oxygen consumption (i.e. ‘flux in’ has increased). By extension, any treatment that increases itaconate will also increase succinate, as competitive inhibition of SDH by itaconate is the means by which SDH is partially inhibited during the pro-inflammatory response.

- Is it possible the breaks happened but oxygen consumption hadn’t changed yet? Are the major enzymes still functioning normally at this point?
 - We have now conducted permeabilized cell respirometry to provide additional evidence that enzymes of the TCA cycle are functioning properly. We offered cells the following substrates to specifically isolate enzymes & pathways potentially affected by cytokine treatment: pyruvate/malate (pyruvate oxidation, specifically PDH); glutamate/malate (glutamate oxidation, specifically α -KGDH), succinate/rotenone (succinate oxidation, specifically SDH), and citrate (IDH). As is seen in **Figs. 4B and 6K**, equal or greater respiratory rates are observed on all substrates with Pam3 or Poly I:C treatment, suggesting (i) that TCA cycle enzymes are functional and (ii) itaconate accumulates via increased production rather than stalling the TCA cycle.
- What are the sources of increased levels of succinate and itaconate?
 - We have conducted stable isotope tracing to look at enrichment of ¹³C₆-glucose and ¹³C₅-glutamine into succinate and itaconate. The results show signal-specific differences in how itaconate is generated (**Fig. 4**): Pam3 treatment increases glucose oxidation and anaplerosis relative to glutamine (as indicated by metabolite enrichment patterns),

whereas Poly I:C treatment shows a relative shift towards glutamine oxidation and anaplerosis relative to glucose. Nonetheless, both treatments show that relative TCA cycle activity is not compromised with either treatment despite itaconate production.

6, A ¹³C-glucose tracing experiment would be highly desirable to demonstrate the contribution of carbon from glucose to the major TCA metabolites including itaconate, and succinate, and particularly citrate, glutamate, fumarate, malate, and aspartate, not negatively influenced at 4h by LPS.

This is an excellent suggestion and lacking from our previous manuscript. We have now conducted stable isotope tracing after 4 hr. treatment with pro-inflammatory stimuli (**Fig. 6**). Again, aligning with our previous data, a mixed MyD88 and interferon signal that collapses oxidative phosphorylation after 24 hr. (i.e. Pam3 + Poly I:C) does not affect incorporation of isotopically labeled glucose into the TCA cycle after 4 hr. Full metabolite abundances and isotopologue distributions are available as a supplementary table associated with the revised manuscript.

7. What is happening to Nos2 gene expression (and more importantly, nitric oxide (NO) levels) in 4h Pam3 – and + PIC cells? If NO amount has increased at this point in time, please provide a discussion on it, particularly why the “breaks” didn’t happen with NO. Supplementary Figure 1 indicates a dramatically increased Nos2 expression at 4h Pam3.

This is an excellent suggestion since nitric oxide is the driver of many mitochondrial alterations observed in pro-inflammatory macrophages. We now include a measurement of the kinetics of *Nos2* gene expression and nitrite accumulation alongside the effect on oxidative phosphorylation in our revised manuscript. As can be seen in **Fig. 6A**, *Nos2* gene expression peaks between 4-5 hrs., at which point several orthogonal measurements of mitochondrial function do not reveal altered energy metabolism or ‘breaks’ in the TCA cycle.

Rather, there appears to be a predictable association between nitrite accumulation in the experimental medium and respiratory inhibition, with profound effects seen between 8-12 hours (**Fig. 6**). The difference in gene expression and nitrite accumulation, of course, likely has to do with the temporal lag between *Nos2* induction, protein synthesis, and activity of the iNOS enzyme. The results reinforce that respiratory inhibition during macrophage pro-inflammatory activation is largely due to iNOS activity, and imply that respiratory inhibition is simply a consequence of *in vitro* NO production and nitrite accumulation rather than a driver of the initial pro-inflammatory macrophage response.

Major points related to Figure 4

8, Figure 4 GHJJ, what was the time of Pam3 stimulation and complex inhibitors (presumably 24h for all)? What were the doses of these complex inhibitors, and the rationale for picking these doses?

We apologize for not making these points clear and organizing the data for the initial submission in a clumsy way, leading to confusion about experimental conditions. The experiments showing that inhibition of the electron transport chain does not enhance Pam3- or Poly I:C-linked gene expression were all conducted for 24 hr. co-treatments with stimuli and respiratory chain inhibitors. We first conducted Seahorse XF analysis to determine the lowest concentration of each inhibitor that resulted in the maximal inhibition of respiration. We erred in not including these important controls in our earlier manuscript – we assumed this was standard best practices and unremarkable – but now include this data as **Fig. S8**.

9, If complex inhibitors were applied for a longer period of time, for example, 24h, please provide cell viability information.

As the reviewer suggests, respiratory chain inhibitors were applied for 24 hr. but did not compromise cell viability. These data are now provided in **Fig. S8D**. Although many cell types cannot survive long-term electron transport chain inhibition, bone marrow-derived macrophages (BMDMs) can tolerate a near-total collapse of oxidative phosphorylation during activation with high concentrations of LPS or Pam3 + Poly I:C. Additionally, previous work has shown that BMDMs can readily meet their energetic demand upon activation with a massive, quantitatively matched increase in glycolysis [Desousa et al. (2023) *EMBO Reports*] to compensate for reduced oxidative phosphorylation. As such, it is expected that respiratory chain inhibition should not cause widespread cell death.

10, If cell viability (cell numbers) were significantly changed, please provide normalized data for Figure 4G.

As indicated in the previous response, cell viability was not altered in response to respiratory chain inhibition over-and-above the growth arrest observed from Pam3 or Poly I:C treatment. All data is also normalized to the housekeeping ribosomal protein 36B4 for each group.

11, Please provide Elisa read outs of major cytokine (IL1 β , IL6 and TNFa) secretion from complex inhibitors + Pam3 treated cells with the same conditions as in Figs. 4G&H, S7A-B, particularly IL1 β .

This is an excellent suggestion, and we now supplement our gene expression data with cytokine secretion levels in **Fig. 5**. The data show that inhibition of the respiratory chain does not alter Pam3-induced cytokine secretion, similarly to the unchanged gene expression.

12, What happened to the Nos2 gene expression in the in vivo macrophages in Figs. 4K&L, and other indicators of an “unbroken” TCA cycle, such as citrate (potentially suc & ita) levels, ACO2/IDH & SDH activity? A Seahorse reading is not strong enough of an evidence to me, as dramatic manipulations have been introduced during the isolation and Seahorse.

This was a very helpful suggestion, and we now include metabolite levels and gene expression from macrophages harvested after intraperitoneal LPS administration in **Fig. 7**. The data show that in addition to normal rates of oxidative phosphorylation, cells show the metabolomic hallmarks of a pro-inflammatory response such as accumulation of succinate, itaconate, and lactate. However, there is no change in citrate abundance, suggesting that nitric oxide levels have not accumulated sufficiently to create extensive damage to the respiratory chain and aconitase/IDH despite an increase in *Nos2* gene expression. The data reinforce the concept that reductions in oxidative phosphorylation and ‘breaks’ in the TCA cycle are not essential for engaging a pro-inflammatory response. Moreover, they suggest the effect of iNOS in *in vitro* cell culture (where nitrite can accumulate in medium over hours) may not accurately reflect an *in vivo* setting where the extracellular environment is consistently circulated and refreshed.

Reviewer #2 (Remarks to the Author):

Major Points

1. Given that previous groups (e.g., Cai et al 2023 JCI; Ran et al 2023 Nat Metab) have shown that aspects of mitochondrial metabolism are dispensable for macrophage activation, it appears the novelty of this manuscript is limited. Primarily, the authors illuminate that the previously characterized 24h timepoint of decreased mitochondrial respiration does not align with the 4h peak pro-inflammatory response following macrophage activation. While significant, the proposed 4h timepoint to investigate the potential interaction of inflammatory gene markers and decreased respiration does not persist in vivo, thus detracting from the significance of the finding altogether.

Prior to the point-by-point response, we would like to thank the reviewer for their thoughtful perspective and suggestions for our manuscript. Addressing these points during our wholesale revision has unquestionably improved our study. We understand the reviewer’s comments about novelty, particularly given the misguided way in which we framed our earlier submission. We initially chose a short-format *Letter* report for clarity, but after the peer-review process it is clear our study required an in-depth, detailed analysis given the interest and previous work in the field. As such, we now present a systematic and rigorous examination of the question using several orthogonal methods along with pharmacologic, genetic, and ex vivo models. Regarding prior work, we note that previous groups have shown some aspects of mitochondrial metabolism can be dispensable for macrophage activation, but note a few points:

- Much of the work in the field stands in direct contrast with other observations. This discrepant data on both sides of the argument can lead to ‘cherry-picking’ of data to support a number of prospective models in the absence of a comprehensive examination of the hypothesis:
 - The manuscripts noted by the reviewers are a good example: Cai et al. (2023) *JCI* shows loss of the NDUFS4 subunit of respiratory complex I can enhance the macrophage inflammatory response (Fig. 1 of Cai et al.). However, Ran et al. (2023) *Nature Metabolism* shows that loss of the mitochondrial pyruvate carrier paralog MPC1 does not affect the

pro-inflammatory response, though this could be explained by a lack of respiratory inhibition due to substrate switching (Fig. 6 of Ran *et al.*). This raises several open questions and possibilities, with the most likely being that respiratory chain activity itself is the trigger for inflammatory macrophage activation. This hypothesis, however, is ultimately disproven with our work.

- Additional genetic evidence is mixed. The loss of NOS2 preserves respiration in macrophages while maintaining a pro-inflammatory response [Palmieri *et al.* (2020) *Nat. Comm.*], but macrophages from *Irg1*^{-/-} macrophages restore respiration and have an attenuated immune response [Lampropoulou *et al.* (2016) *Cell Metabolism*].
- Even studies that claim that mitochondrial ROS is a key driver raise mechanistic inconsistencies. Studies showing the importance of reverse electron transport-mediated ROS [which can be blocked with complex I inhibition; Mills *et al.* (2016) *Cell*] are in conflict with work showing that germline loss of complex I activity itself causes ROS and mimics pro-inflammatory activation [Cai *et al.* (2023) *JCI*].
- In addition to competing data in the literature, it is clear that conventional wisdom has solidified and spread regarding the ‘Warburg-like’ metabolism of ‘M1’ pro-inflammatory macrophages. As the reviewer is no doubt aware, recent high-profile pieces consistently (and incorrectly) propagate the notion that this is an intrinsic feature of pro-inflammatory macrophages in reviews and commentaries [e.g. *Cell Metabolism* (2024) 36(7): 1439-55; *Trends in Endocrinology and Metabolism* (2024) 35(1): 62-73; *Journal of Clinical Investigation* (2023) 33(4):e167079] along with older reviews that continue to be highly cited.
- We ask the reviewer consider our manuscript in light of our choice to submit to *EMBO Reports* and their journal policy on publishing “well-developed, null data on pivotal open questions [and] particularly noteworthy topics...supported by a comprehensive set of negative and positive controls. (<https://www.embopress.org/doi/full/10.15252/embr.202255821>).” We are grateful to the reviewers for encouraging us to conduct a more thorough and exacting study which we hope meets this mark.

As a broad point of comparison, our understanding of cancer cell metabolism has undergone a dramatic shift in the past decade. It has evolved from a one-size-fits-all, ‘Warburg-like aerobic glycolysis’ phenotype to an appreciation of the role of oxidative phosphorylation and metabolic plasticity in supporting oncogenesis. We hope our report helps support a similarly evolved discussion about the nuance, plasticity, and spectrum of phenotypes accompanying macrophage polarization.

2. As the authors point out, the lack of contiguity between in vitro and in vivo bioenergetic inflammatory macrophage phenotype leaves much to be desired. Notably, previous work has demonstrated that BMDMs and tissue-resident macrophages often have different (or unique) bioenergetics. Given that the bulk of the findings in the current manuscript were acquired in BMDMs, it stands to reason that these experiments should be replicated in peritoneal macrophages (as an example, given the authors also used these cells in the current manuscript), which could identify specific mechanistic steps responsible for the lack of continuity between the in vitro and in vivo findings.

This is an important, constructive comment and we agree that pinpointing the mechanisms responsible for the lack of continuity between the *in vitro* and *in vivo* findings would be meaningful. We now present additional data in **Fig. 5** (NO-generation time-course aligned with gene expression and respiratory inhibition) & **Fig. 7** (further *ex vivo* analysis of *in vivo*-activated peritoneal macrophages) towards this end. As we note in the discussion, the data suggest that many of the *in vitro* results from BMDMs that have shaped dogma in the field are likely passive consequences of *in vitro* nitrite accumulation in cell culture dishes rather than requisite signals that drive macrophage function.

3. The authors highlight previous reports of the TCA “break” to say that if macrophage activation leads to itaconate and succinate accumulation, the cells would not be able to fuel OXPHOS. But in their experiments, macrophages produce itaconate and succinate and still effectively enter OXPHOS, leading them to claim this finding is sufficient to conclude that macrophage activation does not decrease mitochondrial respiration

through a TCA break, driving itaconate/succinate production instead of aKG/NADH. This conclusion discounts the role of ROS production in both macrophage activation and mitochondrial respiration. To support this conclusion, the authors need to measure aKG/NADH/ROS together with measurements of itaconate/succinate. Otherwise, one cannot know if there is just higher overall metabolism or if macrophages are actually diverting to one pathway or another. Stable isotope tracing of exogenous pyruvate and/or isocitrate could also address this point.

We thank the reviewer for this comment that helped refocus and refine our manuscript. We would like to direct the reviewer to comment #5 from Reviewer #1, as many of these points are addressed. Some additional points are also worth noting:

- We have conducted stable isotope tracing with uniformly labeled $^{13}\text{C}_6$ -glucose and $^{13}\text{C}_5$ -glutamine after both 4 hr. and 24 hr. treatment of BMDMs with various pro-inflammatory stimuli. This data generates information about both the steady-state abundance of metabolites and the enrichment from labeled substrates. All data (metabolite abundances and mass isopologue distributions from isotopic labels) are now available in the supplementary materials.
- We now do a better job in both the introduction and the conclusion of explicitly stating that our findings do not exclude mitochondrial signals (e.g. mitochondrial ROS from reverse electron transport) that are compatible with healthy oxidative phosphorylation. We do not claim – nor does the data support – that there is no role for mitochondrial function *per se* in the pro-inflammatory response. Rather, our experiments were designed to test the conventional model that the generation of any putative signals requires the inhibition of oxidative phosphorylation. We thank the reviewer for helping reveal this blind spot in how we framed our initial submission.

*4. According to the title and manuscript narrative, decreased mitochondrial respiration has (controversially) been thought to precede macrophage activation. Activated macrophages, though, decrease oxidative phosphorylation and induce accumulation of mitochondrial metabolites, suggesting a bidirectional link between macrophage activation and mitochondrial respiration. Although the authors demonstrate that LPS and combined Pam3 + Poly I:C reduce mitochondrial respiration, the inability to alter the pro-inflammatory landscape with myeloid-specific loss of *Ndufs4* subunit of complex I, as shown in previous studies, leaves a significant gap in the understanding of respiratory chain dysfunction by this model. Pharmacologic inhibition of the respiratory chain of mitochondrial metabolism alone is an incomplete perturbation of pro-inflammatory macrophage activation. Although this Reviewer understands that genetic deletion often leads to compensation, especially when targeting metabolic enzymes and transporters, there are numerous other approaches the authors could use to address this, such as shRNA enzyme targeting and/or enzyme overexpression, especially in combination with the experiments suggested in Point 3.*

This was an excellent suggestion and we thank the reviewer for suggesting this experiment. The addition of this genetic proof-of-concept has strengthened our conclusions and the underlying rigor of the experiments. We now include CRISPR-mediated deletion of *Ndufs4* in J2-immortalized primary BMDMs in our manuscript (**Fig. 5** of the revised manuscript) and couple our prior pharmacologic data with genetic inhibition of respiratory chain activity. The results show that acute loss of *Ndufs4* in immortalized macrophages does not enhance the pro-inflammatory response in this model. This contrasts the effects observed in myeloid-specific, germline loss of the protein and the implications of this finding are addressed in the revised manuscript discussion.

Minor Points:

1. The authors claim in Figure 1B that the 4h timepoint is “sufficient to induce inflammatory cytokine release.” However, they should also show the 24h timepoint, as in the other panels of Figure 1 as a thorough source of comparison. If, in fact, the 24h timepoint shows significantly higher levels of IL-6 and/or IL-12/IL-23, the explanation of Figure 1B should be stated with softer language.

We very much appreciate the suggestion to use more appropriate language around the time-course experiment and have done so in our thoroughly reorganized and expanded manuscript. Regarding the cytokine release assays, we note that these measurements are generally end-point assays and, as such, the 24 hr. timepoint will necessarily have a greater reading than at 4 hr. A rate-based cytokine release assay could be informative, but it is likely subject to artifacts of low signal-to-noise on an hour-to-hour basis to determine when the maximal rate of cytokine release occurs.

2. Similarly, the levels of lactate efflux following 4h and 24h with activated by LPS (bar 2 and 4 in Figure 1C) appear by eye to differ significantly. Statistical comparisons between these bars should be shown, particularly since the 24h timepoint, not the 4h timepoint, is used in Figure 4M to assess ex vivo peritoneal macrophage lactate efflux. Combined, this seems to suggest that in fact 4h may not induce sufficient lactate efflux post-LPS treatment.

We thank the reviewer for their comment to include a more formal analysis of glycolysis in our manuscript. As such, we now provide an analysis showing signal-specific regulation of glycolysis using genetic and pharmacologic models in Supplemental **Fig. S3**, which reveal the substantial increase in glycolysis upon inflammatory macrophage activation is driven largely by MyD88. In addition to this, previous work (PMID: 37548091) provides a quantitative, rigorous analysis of the time course and signal specificity of LPS-activated macrophages. This work also shows that oxidative phosphorylation and glycolysis are discretely regulated upon macrophage activation.

We agree with the specific point that the lactate efflux rate is likely not maximal after 4 hr. and may not be as high as after 24 hr. of stimulus. We did a poor job of explaining that the rationale for the measurement was to have additional data alongside gene expression to show that the pro-inflammatory response had been triggered after 4 hr. In this context, the many-fold increase in glycolytic rate (even if not maximal) further reinforces that an ample pro-inflammatory response has been triggered after 4 hr. We again apologize for the confusion stemming from how awkwardly our initial submission was framed & presented.

3. Statistical analysis used in Figure 3K is not stated.

Unfortunately, we were only able to obtain two distinct human samples from different donors, so the data are presented here as mean +/- spread of biological replicates as indicated in the figure legend. Although quantitative statistics cannot be applied, the results nonetheless demonstrate an unequivocal increase in itaconate production upon activation of human monocyte-derived macrophages.

Reviewer #3 (Remarks to the Author):

1. One line of support presented in your data suggesting that suppressing respiration is not necessary for pro-inflammatory activation is due to LPS-induced cytokine production at 4hrs despite respiration being suppressed only after 24hrs. However, it seems that a Myd88 + TRIF-mediated inflammatory response can be segregated into 2 phases, an early and a late phase, that may have distinct mechanisms of action, with the later phase maybe requiring a suppression of respiration to maintain a pro-inflammatory phenotype. Would suppressing respiration at 4hrs have a different effect on cytokine production compared to 24hr? (what time point is used for Figure 4G,H?)

This was an excellent suggestion to further test the underlying biology of whether oxidative phosphorylation can regulate the inflammatory response. In our expanded and thoroughly reworked manuscript, we now show that even 4 hr. after cytokine treatment, respiratory inhibition has no effect on pro-inflammatory gene expression and cytokine release (**Fig. 6**). We have also conducted additional measurements of NADH fluorescence lifetime, ¹³C stable isotope tracing, and mitochondrial membrane potential to better understand the bioenergetic response to short-term (4 hr.) pro-inflammatory signaling.

2. The compounds used such as LPS or poly I:C are useful to tease out signaling by specific TLRs and signaling pathways. However, could you make similar conclusions with a pathological pro-inflammatory signal such as bacterial infection, oxidized lipids, etc? Could be explored both in vitro and in the in vivo model. Direct injection of LPS or Pam3 intraperitoneally seems quite limiting.

This is an excellent suggestion and we now show *Staphylococcus aureus*, a physiologically relevant TLR2 ligand, elicits the same metabolic profile at a synthetic TLR2 agonist, Pam 3 (**Figs. S1 & S2**).

3. You focus on the the changes to succinate and itaconate. However, it seems that there is an interesting response in citrate levels that lines up with the changes in respiration. Citrate is unchanged at 4hr with any combination of Pam3 and poly I:C treatment, though it becomes greatly responsive only to a combination of Pam3 and poly I:C at 24hrs correlating with the suppression of respiration. Citrate can by shuttled into the

cytoplasm to enhance histone acetylation following ATP-citrate lyase activity contributing to the pro-inflammatory response (Lauterbach et al Immunity 2019 - one of your citations, nitric oxide induced citrate accumulation - Palmieri et al Nat Comm 2020, also in your citation list). The effect on citrate could be looked into a bit further.

We thank the reviewer for this thoughtful and knowledgeable suggestion. Given our expanded manuscript now includes permeabilized cell respirometry and stable isotope tracing (**Figs. 4, 6, and 7**), we have data to support that the accumulation of citrate is simply a result of NO-mediated inhibition of isocitrate dehydrogenase. It would be very interesting, indeed, to examine whether this increase in steady-state citrate abundance leads to increased ACLY activation and potentially alters histone acetylation. This is particularly true when also considering that MyD88-linked *de novo* lipogenesis (PMID: 32516576) also requires large amounts of mitochondrial citrate efflux and acetyl CoA. However, given that the increase in citrate does not correlate with pro-inflammatory macrophage activation *per se*, a more thorough examination is likely beyond the scope of testing the present hypothesis.

4. How do your treatments to manipulate respiration affect glycolysis? Glycolytic metabolism is a crucial part of pro-inflammatory macrophage activation and changes in glycolysis could be a confounding factor in some of your findings.

We thank the reviewer for bringing up this important point given the well-established increase in glycolysis upon pro-inflammatory macrophage activation. We now show signal-specific regulation of glycolysis using genetic and pharmacologic models in Supplemental **Fig. S3**, which shows the substantial increase in glycolysis upon inflammatory macrophage activation is driven largely by MyD88. Additionally, previous work (PMID: 37548091) has shown that oxidative phosphorylation and glycolysis are discretely regulated upon macrophage activation.

Were we to see that manipulating mitochondrial function had an attenuating effect on the pro-inflammatory response, then indeed it may be that glycolysis could not meet the ATP demand and our results could not discriminate between a direct effect of mitochondrial signaling or an indirect effect from dying, non-functional cells. However, showing no link between respiratory inhibition and macrophage activation suggests bioenergetic plasticity in the metabolic phenotypes that can accommodate effector function. We now do a better job of addressing this point in our revised manuscript.

5. How does blocking the suppression of respiration affect the expression of pro-resolving/anti-inflammatory cytokines (e.g. IL10)? With extended pro-inflammatory signaling (e.g. LPS), macrophages eventually transition to more of a resolution phenotype. How do your manipulations of respiration affect the expression of pro-resolving cytokines? Suppression of respiration has been found to prevent this transition previously.

How mitochondrial function affects the resolution of the pro-inflammatory response is a fascinating topic. Given the scope of our current study, however, we believe an examination of resolution (which requires a thorough and systematic analysis beyond *IL10* expression) is best left to an independent follow-on study focused solely on that subject.

6. Your outputs focus on cytokine production and succinate + itaconate accumulation. Pro-inflammatory macrophages are also defined by their phagocytic capacity and other critical behaviors. Do your conclusions also hold for other functions of pro-inflammatory macrophages?

We again thank the reviewer for this helpful suggestion that has improved the rigor of our findings. We have incorporated functional measurements to now show that inhibition of oxidative phosphorylation has no effect (or even a slightly inhibitory effect) on phagocytosis (**Fig. 5J**).

In addition, just a few minor points:

1. Check the y-axis numbering for some of the graphs such as figure 1A and 1B, 4A and 4G for IL12b mRNA graphs.

We apologize for this error and thank the reviewer for their attention to detail. The manuscript has now been edited accordingly.

2. In figure 2A, maximum respiration seems to be lowered by half following Pam3 treatment only compared to

ctrl or poly I:C only. However in the quantification in figure 2B, maximum respiration with Pam3 treatment does not seem different from poly I:C treatment.

We thank the reviewer for their thoughtful analysis of the results. The discrepancy comes from the presentation of a single representative trace (one cell preparation from one animal presented with technical replicates; Fig. 2A) and the aggregate data from several biological replicates derived from distinct animals and conducted on different days (2B).

3. Some figures in main and supplementals missing length of treatment (4 hr? 24hr?), should include in the legend.

We apologize for the confusion and realize our previous sequence of data presentation was clumsy and incomplete. The manuscript has now been edited accordingly to explicitly note the time frame and concentrations of all treatments.

Dear Ajit,

Thank you for the transfer of your revised manuscript to EMBO reports. We have now received the enclosed reports from the referees and I am happy to say that both support its publication by EMBO reports.

Only a few editorial requests will need to be addressed before we can proceed with the official acceptance of your manuscript.

- Please add up to 5 keywords to the ms file.
- Please add a "Data Availability Section" to the final ms that lists links and accession numbers for primary datasets produced in this study and deposited in public databases. If your study has not produced and deposited novel datasets, please mention this fact in the Data Availability Section.
- Please correct the conflict of interest subheading to "Disclosure Statement and Competing Interests"
- Please remove the author credits from the ms file. All credits need to be entered during online ms submission.
- Please correct the REFERENCE FORMAT to the EMBO reports (Harvard) style - it needs to be alphabetical, not numerical; et al needs to be used after 10 author names; 3 references have DOIs but DOIs should only be used for preprints and datasets that have not been published yet.
- Please co-submit a completed author checklist, which you can download from our author guidelines <<https://www.embopress.org/page/journal/14693178/authorguide>>. The completed author checklist will also be part of the transparent peer-review process file.
- The heading of the FUNDING INFO should be removed as this info needs to be part of the Acknowledgments. 4814 grant number (of Agilent Technologies (Agilent) funder) is missing in the ms file, please correct.
- A callout for Fig. 5O is missing, please add.
- The title of the Supplementary Materials file should be "Appendix"; we need a table of content with each Appendix item listed and its page number on the first page; the nomenclature throughout the file should be corrected from Supplementary Figure 1, etc. to Appendix Figure S1, etc. from Table S1 to Appendix Table S1; ms callouts of these figures and table need to be updated accordingly.
- The Methods section needs to include a Reagents and Tools Table (listing key reagents, experimental models, software and relevant equipment and including their sources and relevant identifiers) as a separate file. A downloadable template (.docx) for the Reagents and Tools Table can be found in our author guidelines: <<https://www.embopress.org/page/journal/14693178/authorguide#manuscriptpreparation>>.
- The Abbreviations section needs to be removed from the manuscript. Abbreviations should be defined in brackets after their first mention in the text, not in a list of abbreviations.
- Please define the annotated p values ****/**/*/* as well as provide the exact p-values for the same in the legend of figure 1C, F, G, K; 2B, C, D, F, G, H; 3B, C, D, F; 4A, B, D, E; 5C, D, E, J, N; 6A, B, C, I; J, L; 7B-F, I as appropriate.
- Please indicate the statistical test used for data analysis in the legends of figures 1K, 6B, E, G, I.
- Please note that scale bar and its definition are missing for figures 1J; 6H.
- Please note that axis labels are not defined for figures 1C, F, G, I, K; 2D, F; 3B, C, G, H; 4B, 5A, B; 6E, J, K, L; 7B, C, D, E, F, H, I;
- Please note that axis gaps are not labeled appropriately in figures 1K.

EMBO press papers are accompanied online by A) a short (1-2 sentences) summary of the findings and their significance, B) 2-3 bullet points highlighting key results and C) a synopsis image that is exactly 550 pixels wide and 200-600 pixels high (the height is variable). The synopsis image should provide a sketch of the major findings, like a graphical abstract. Please note that text needs to be readable at the final size. Please send us this information along with the final manuscript.

Best regards,

Esther

Referee #1:

I have reviewed the revised manuscripts and I am satisfied that the authors have sufficiently addressed the comments I raised previously.

Referee #2:

The authors have more than adequately addressed my concerns and merits publication at this stage.

All editorial and formatting issues were resolved by the authors.

Ajit Divakaruni
University of California, Los Angeles
Department of Molecular and Medical Pharmacology;
650 Charles E Young Dr. S
Los Angeles, CA 90095
United States

Dear Ajit,

I am very pleased to accept your manuscript for publication in the next available issue of EMBO reports. Thank you for your contribution to our journal.
